# Microtubule asters anchored by FSD1 control axoneme assembly and ciliogenesis

Hai-Qing Tu[1], Xuan-He Qin[1], Zhi-Bin Liu[2,3], Zeng-Qing Song[1], Huai-Bin Hu[1], Yu-Cheng Zhang[1], Yan Chang[1], Min Wu[1], Yan Huang[1], Yun-Feng Bai[1], Guang Wang[1], Qiu-Ying Han[1], Ai-Ling Li[1], Tao Zhou[1], Feng Liu [ID] [3], Xue-Min Zhang [ID] [1] & Hui-Yan Li[1,4]

Defective ciliogenesis causes human developmental diseases termed ciliopathies. Microtubule (MT) asters originating from centrosomes in mitosis ensure the fidelity of cell division by positioning the spindle apparatus. However, the function of microtubule asters in interphase remains largely unknown. Here, we reveal an essential role of MT asters in transition zone (TZ) assembly during ciliogenesis. We demonstrate that the centrosome protein FSD1, whose biological function is largely unknown, anchors MT asters to interphase centrosomes by binding to microtubules. FSD1 knockdown causes defective ciliogenesis and affects embryonic development in vertebrates. We further show that disruption of MT aster anchorage by depleting FSD1 or other known anchoring proteins delocalizes the TZ assembly factor Cep290 from centriolar satellites, and causes TZ assembly defects. Thus, our study establishes FSD1 as a MT aster anchorage protein and reveals an important function of MT asters anchored by FSD1 in TZ assembly during ciliogenesis.

[1] State Key Laboratory of Proteomics, National Center of Biomedical Analysis, Beijing 100850, China. [2] University of Chinese Academy of Science, Beijing 100101, China. [3] State Key Laboratory of Biomembrane and Membrane Biotechnology, Institute of Zoology, Chinese Academy of Sciences, Beijing 100101, China. [4] Cancer Research Institute of Jilin University, The First Hospital of Jilin University, Changchun, Jilin 130021, China. These authors contributed equally: Hai-Qing Tu, Xuan-He Qin, Zhi-Bin Liu. Correspondence and requests for materials should be addressed to F.L. (email: liuf@ioz.ac.cn) or to X.-M.Z. (email: zhangxuemin@cashq.ac.cn) or to H.-Y.L. (email: hyli@ncba.ac.cn)

The primary cilia are immotile microtubule (MT)-based structures present in most types of mammalian cells[1,2]. This antenna-like organelle plays an important role during embryonic development by integrating extracellular signals[3–5]. Defects in ciliary function have been linked to a number of human diseases termed ciliopathies, including Bardet-Biedl syndrome, Meckel syndrome, nephronophthisis, and polycystic kidney disease[6–9]. In non-dividing cells, the mother centriole acts as the basal body to template the formation of cilia[10,11]. Unlike the daughter centriole, the mother centriole is specifically characterized by distal and subdistal appendages. The distal appendages dock the centriole to the plasma membrane during ciliogenesis[12,13], whereas the subdistal appendages anchor cytoplasmic MT asters in interphase[14].

In interphase cells, MT asters tethered to the subdistal region of mother centrioles through their minus ends influence cell polarity and cell motility[15,16]. Several subdistal appendage proteins, such as Ninein, ODF2, Kif3a, and p150$^{Glued}$, have been reported to regulate MT aster anchoring[17,18]. As a component of the subdistal appendage, Ninein has been reported as a major regulator in anchoring MT asters at this structure[19]. It is known that other regions, such as the pericentriolar material (PCM), also play a critical role in MT anchoring[20]. Other proteins, including EB1, SSX2IP, Trichoplein, and CAP350/FOP/CEP19, which are close to subdistal appendages of mother centrioles, have also been shown to participate in MT aster anchoring possibly through Ninein[21–26]. Interestingly, all of these known anchoring proteins are also required for ciliogenesis[27], suggesting that MT asters contribute to cilia assembly. There, however, no data demonstrating that these MT anchoring proteins promote ciliogenesis through their anchoring activity. How MT asters regulate cilia assembly also remains unknown.

During ciliogenesis, the axoneme grows vertically from the mother centriole through a conserved multi-step process and consists of nine doublet MTs[28,29]. The transition zone (TZ) assembly is a critical step for the growth of nine doublet MTs in the axoneme[30,31]. The TZ is a subcompartment situated at the base of ciliary axoneme and acts as a gate of protein diffusion to control ciliary composition and signaling[32,33]. The importance of TZ is highlighted by the finding that many human ciliopathies are caused by mutations of various TZ components, including the MKS and NPHP modules[34].

Centriolar satellites (CSs) are electron-dense structures that accumulate in the vicinity of centrosomes in a manner depending on dynamic MTs. It has been reported that several satellite proteins, including PCM1, Talpid3, Cep72, and SSX2IP, participate in ciliogenesis at least in part through Cep290[35–38]. Cep290 localizes to both centriolar satellites and centrioles prior to cilia formation and plays multiple essential roles in ciliogenesis, including TZ assembly[30,33,39,40]. How Cep290 specifically contributes to TZ assembly remains unclear.

FSD1 (fibronectin type III and SPRY domain containing 1, also known as MIR1 and GLFND) is originally identified as a centrosome/MT-associated protein[41,42]. However, its biological function remains unknown. In this study, we report that FSD1 controls TZ assembly during ciliogenesis. We found that FSD1 localizes to the middle region of mother centrioles, directly binds to MTs, and anchors MT asters at centrosomes. Inactivation of FSD1 or other anchoring proteins disrupts anchoring of MT asters and the dynamic localization of Cep290 at centriolar satellites in interphase, resulting in TZ assembly defects during ciliogenesis. Thus, our data reveal an important function of the anchoring of MT asters at interphase centrosomes for subsequent ciliogenesis.

## Results

**Depletion of FSD1 blocks ciliogenesis in mammalian cells.** Previous reports have shown that FSD1 is a centrosome/MT-associated protein, but its function remains unknown[41,42]. We found that FSD1 co-localizes with γ-tubulin, a canonical centrosome marker, in human retinal pigment epithelial cells (RPE-1) at different stages of the cell cycle (Supplementary Fig. 1a). We performed immunofluorescence experiments with a mouse monoclonal antibody against FSD1. The transfection of an short interfering RNA (siRNA) against FSD1 abolished FSD1 staining at centrosomes, demonstrating the specificity of the antibody (Supplementary Fig. 1b). The centrosome localization of FSD1 and the specificity of the antibody were also confirmed in U2OS cells (Supplementary Fig. 1c).

When cells exit the cell cycle, the mother centriole within the centrosome forms the basal body, which initiates cilia assembly[28,43–45]. We next investigated the possible function of FSD1 in ciliogenesis. We found that depletion of FSD1 with siRNA caused a dramatic reduction in primary cilia assembly in RPE-1 cells after serum starvation (Fig. 1a). We further confirmed this defect in cilia formation using three individual siRNAs against FSD1 by staining cells with antibodies against two ciliary markers, acetylated α-tubulin (Ac-tubulin) or ADP ribosylation factor like GTPase 13B (ARL13B) (Fig. 1b and Supplementary Fig. 1d–g). Importantly, the defect of ciliogenesis induced by FSD1 knockdown was rescued by expressing an siRNA-resistant form of FSD1 (Fig. 1c). The levels of ectopically expressed FSD1 proteins were comparable to that of the endogenous FSD1 (Supplementary Fig. 1h). In addition, the knockdown of FSD1 did not decrease the percentage of quiescent cells, as determined by staining with the proliferation marker Ki-67 (Supplementary Fig. 1i, j). This result rules out the possibility that depletion of FSD1 might affect exit from the cell cycle and indirectly cause ciliogenesis defects. The siRNA #1 was used in all of the FSD1 RNAi experiments in this study unless indicated otherwise. Taken together, these data suggested that FSD1 is required for cilia formation in mammalian cells.

**Fsd1 modulates ciliogenesis and cilia function in zebrafish.** In zebrafish, ciliary dysfunction can be discerned by defects in left-right (LR) asymmetry[46,47]. To further test the importance of fsd1 in ciliary function in zebrafish, we designed two distinct antisense morpholinos (MOs) to knockdown fsd1: aMO blocks translation, whereas sMO blocks splicing. The knockdown efficiency of fsd1-aMO was validated by western blot (Supplementary Fig. 2a). The knockdown efficiency of fsd1-sMO was analyzed by sequencing and reverse transcription-PCR (Supplementary Fig. 2b). In zebrafish, the knockdown of fsd1 induced by aMO or sMO results in aberrant embryo development. In fsd1-knockdown morphants, more than 60% embryos had curved body and 70% embryos had pericardial edema at 72 h post fertilization (hpf) with a dose-dependent severity (Fig. 1d, e). Notably, other morphological defects, such as defective LR asymmetry, were also observed in fsd1-knockdown morphants in a dose-dependent manner as revealed by whole-mount in situ hybridization. spaw, a marker of left lateral plate mesoderm[48], distributes mainly to the left side at the 18-somite stage in control zebrafish (Fig. 1f). However, more than 50% of embryos injected with fsd1-aMO or sMO displayed bilateral and even right-sided expression of spaw in a dose-dependent manner (Fig. 1f, g). As a marker for cardiac mesoderm, the expression of cmlc is normally left-sided at 26 hpf[49]. Fsd1-knockdown embryos displayed middle- and even right-sided expression of cmlc in a dose-dependent manner (Supplementary Fig. 2c). In addition, we found that the LR asymmetry

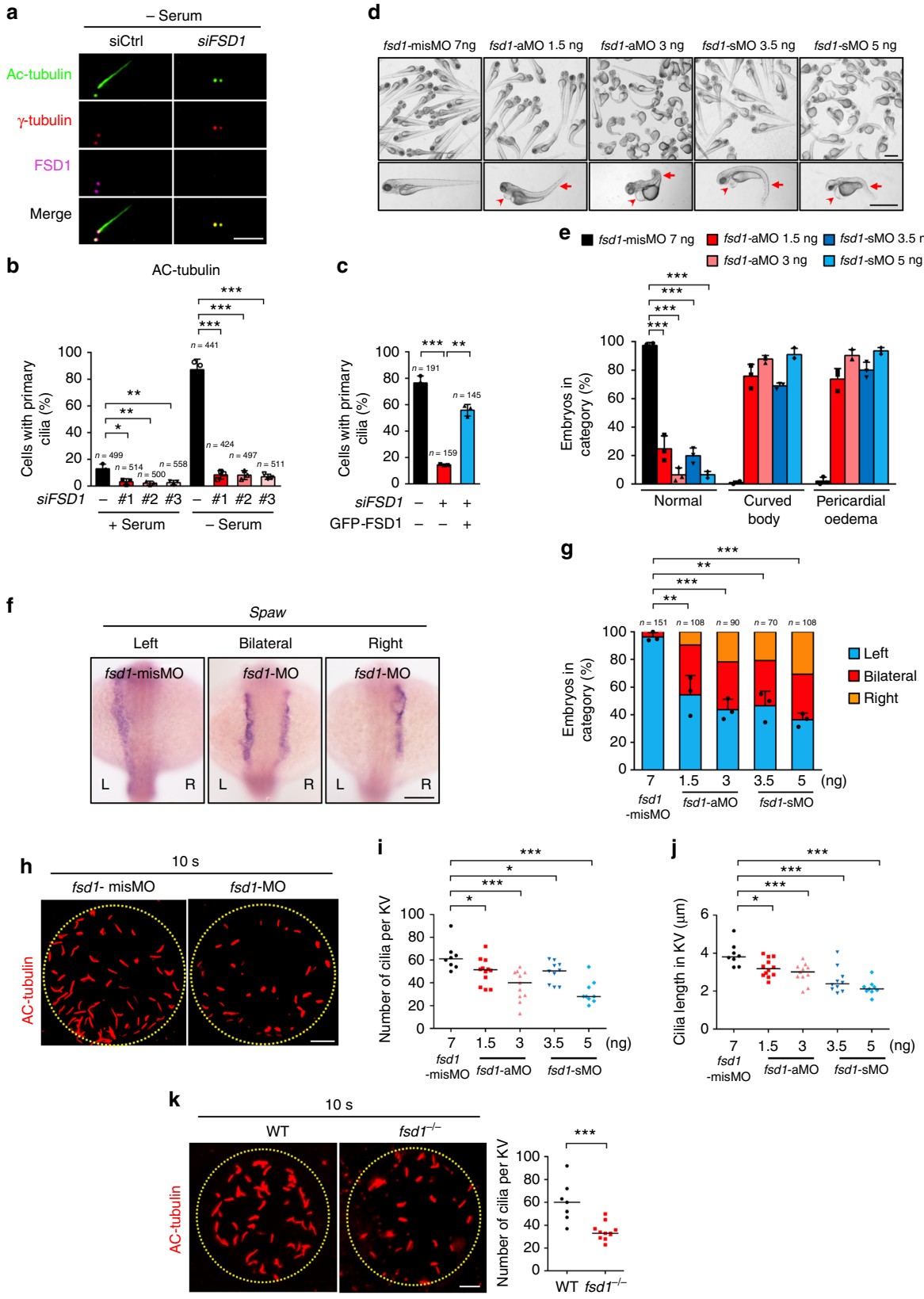

defects induced by *fsd1*-aMO could be partially rescued by expressing an aMO-resistant form of zebrafish *fsd1* mRNA (*fsd1* re-zmRNA) (Supplementary Fig. 2d–f). In contrast, a synthesized mutant mRNA encoding a truncated fsd1 protein (1–119 aa) (*fsd1*-CC re-zmRNA) could not rescue the LR asymmetry defects

in *fsd1* morphants (Supplementary Fig. 2g–i). Taken together, these data suggested that fsd1 is required for the LR asymmetry in zebrafish.

As LR asymmetry is controlled by ciliary function in Kupffer's vesicle (KV)[50], we next counted the cilia number and measured

**Fig. 1** FSD1 is required for ciliogenesis in human cells and zebrafish. **a** RPE-1 cells were transfected with the indicated siRNA and serum-starved for 48 h. Cells were stained with the indicated antibodies and examined by immunofluorescence microscopy. Scale bar, 5 μm. **b** Effects of FSD1 depletion on cilia formation in cycling (+Serum) or quiescent (−Serum) RPE-1 cells. Acetylated α-tubulin (Ac-tubulin) is ciliary marker. **c** Cilia defects induced by FSD1 knockdown in RPE-1 cells were rescued by expressing a GFP-tagged, RNAi-resistant form of FSD1. Data are presented as mean ± s.d. of three independent experiments. *n* number of cells. **d**, **e** *Fsd1* morphants (aMO and sMO) displayed curved body and pericardial edema at 72 hpf. The arrows mark curved body and arrowheads mark pericardial edema. The *fsd1*-misMO were used as control. Scale bars, 1 mm. Data are presented as the mean ± s.d. of three independent experiments. **f**, **g** *Fsd1* MOs (aMO and sMO) caused left-right asymmetry defects. The *spaw* probe was used to label the left lateral plate mesoderm in the whole-mount in situ hybridization at 18-somite stage. Scale bar, 150 μm. *n* number of fishes. **h–j** *Fsd1* MOs (aMO and sMO) impaired ciliogenesis in Kupffer's vesicle at 10-somite stage (10 s). Bars indicate the median. Scale bar, 10 μm. **k** *Fsd1* knockout impaired ciliogenesis in Kupffer's vesicle at 10 s. Bars indicate the median. Scale bar, 10 μm. In all panels, statistical comparisons between two groups were carried out by two-tailed *t*-test. *$P < 0.05$, **$P < 0.01$, ***$P < 0.001$

the cilia length in KV at 10-somite stage. Our results showed that knockdown of *fsd1* significantly reduced both the number and length of cilia in KV (Fig. 1h–j). In addition, we used CRISPR-Cas9 to knockout the *fsd1* gene in zebrafish and observed significant cilia-associated defects (Supplementary Fig. 3a), including ciliogenesis defect (Fig. 1k), curved body, and LR asymmetry defect (Supplementary Fig. 3b, c). Collectively, our results indicate that fsd1 is required for proper ciliogenesis and ciliary function during embryonic development.

**Loss of FSD1 blocks ciliogenesis at the TZ assembly stage**. In vertebrate cells, cilia biogenesis consists of a series of conserved steps, including centriole maturation into a basal body, docking of the basal body to the plasma membrane, and extension of the axoneme (Supplementary Fig. 4). To investigate how FSD1 regulates ciliogenesis, we tested the effect of FSD1 knockdown on the localization of known complexes that play key roles in various steps of ciliogenesis. In FSD1-depleted cells, γ-tubulin, and Pericentrin localized normally to centrosomes, indicating that FSD1 is dispensable for the centrosome integrity (Fig. 2a and Supplementary Fig. 5a–c). Next, we examined the effect of FSD1 knockdown on the ciliary vesicle (CV) formation. In FSD1-depleted cells, several proteins with known functions in CV formation, including Cep164, IFT20 and Rab8a, maintained their regular localization at the ciliary base (Fig. 2a, b and Supplementary Fig. 5a, d, e), suggesting that FSD1 might not be required for CV formation. Moreover, two key negative regulators of ciliogenesis[51], CP110 and Cep97 disappeared normally from mother centrioles in the absence of FSD1 in quiescent cells (Fig. 2a, c and Supplementary Fig. 5f). We also found that FSD1 knockdown did not affect the localization of TCTN1 and MKS1 (two non-membrane proteins of the MKS module), two TZ proteins, which constitutively localize at the centriole during ciliogenesis (Fig. 2a and Supplementary Fig. 5a, g, h). Taken together, our results support a role of FSD1 during early steps of ciliogenesis.

We then examined the recruitment of several TZ proteins, including components of the MKS module[33,52,53] and the NPHP1-4-8 complex[34,39]. Interestingly, FSD1 depletion severely disrupted the recruitment of TMEM67 and TCTN2 (Fig. 2a, d and Supplementary Fig. 5a, i), two transmembrane components of the MKS module[39,53]. In addition, two components of the NPHP1-4-8 complex, NPHP4 and NPHP8 failed to be recruited to the basal body in FSD1-depleted cells (Fig. 2a, e and Supplementary Fig. 5a, j). Therefore, these data demonstrate that FSD1 facilitates the recruitment of TZ components to the TZ (Fig. 2a and Supplementary Fig. 5a).

To confirm whether TZ assembly is impaired in FSD1-depleted RPE-1 cells, we examined the structure of TZ using transmission electron microscopy. As expected, in control cells, more than 80% (32/38) of mature centrioles docked onto vesicle membranes and supported the formation of axoneme known to organize the TZ

(Fig. 2f, left). In contrast, an intact and flattened CV docked at the distal ends of mother centriole in 90% (28/30) of FSD1-depleted cells (Fig. 2f, right). There was also evident outgrowth of a short ciliary bud, but this short bud failed to deform the CV membrane (Fig. 2f, right), indicating a lack of intact TZ structure. Taken together, our results demonstrate a requirement of FSD1 for the assembly of TZ during ciliogenesis.

**FSD1 forms a ring structure at the centriolar middle region**. To further characterize the mechanism by which FSD1 promotes TZ assembly, we tested whether FSD1 is a TZ component. We defined the precise localization of FSD1 within the basal body using several centriolar markers. We found that FSD1 localized between C-Nap1 and Cep162, markers of centriole proximal ends and distal ends, respectively (Fig. 3a, b). Intriguingly, FSD1 staining overlapped with that of subdistal appendage protein ODF2, which marks the subdistal region of centrioles, but FSD1 was completely separated from centriole distal end markers, including Centrin2, IFT88, Cep162, and distal appendage marker Cep164 (Fig. 3a, b). We next tested whether FSD1 could localize to the TZ during ciliogenesis. Unexpectedly, we found that FSD1 localized beneath Cep162 during ciliogenesis (Fig. 3c). FSD1 staining did not overlap with that of MKS1 and NPHP8, which mark the TZ of primary cilia (Fig. 3d). Taken together, our data show that FSD1 constitutively localizes to the middle region of both centrioles in cycling and quiescent cells.

To further define the localization of FSD1, we obtained super-resolution immunofluorescence images using the stimulated emission depletion microscopy (STED). Remarkably, these images showed that FSD1 formed a ring structure encircling Centrin2 (Fig. 3e). The FSD1 ring did not overlap with Cep164 and FBF1, which mark the distal appendage of mother centrioles (Fig. 3e). Thus, our results indicate that FSD1 does not localize to the TZ, but rather forms a ring structure at the middle region of both centrioles (Fig. 3f).

**The maintenance of Cep290 at CS is mediated by FSD1**. To clarify how FSD1 facilitated the recruitment of TZ components to the mother centriole, we examined its relationship with Cep290 and Cep162, which had been reported to promote TZ assembly[52,54]. We found that loss of FSD1 did not affect the localization of Cep290 and Cep162 at both centrioles in serum-starved RPE-1 cells (Supplementary Fig. 6a, b). Because Cep290 also localizes to CS before the formation of cilia[35], we hypothesized that this population of Cep290 might participate in TZ assembly during ciliogenesis.

To test this idea, we detected the effect of FSD1 knockdown on the localization of Cep290 at CS. Consistent with previously reported results[35], we indeed found that CS Cep290 disappeared after serum starvation, and the Cep290 granules at CS negatively correlated with cilia assembly (Fig. 4a, Supplementary Fig. 6c). Strikingly, knockdown of FSD1 specifically affected the

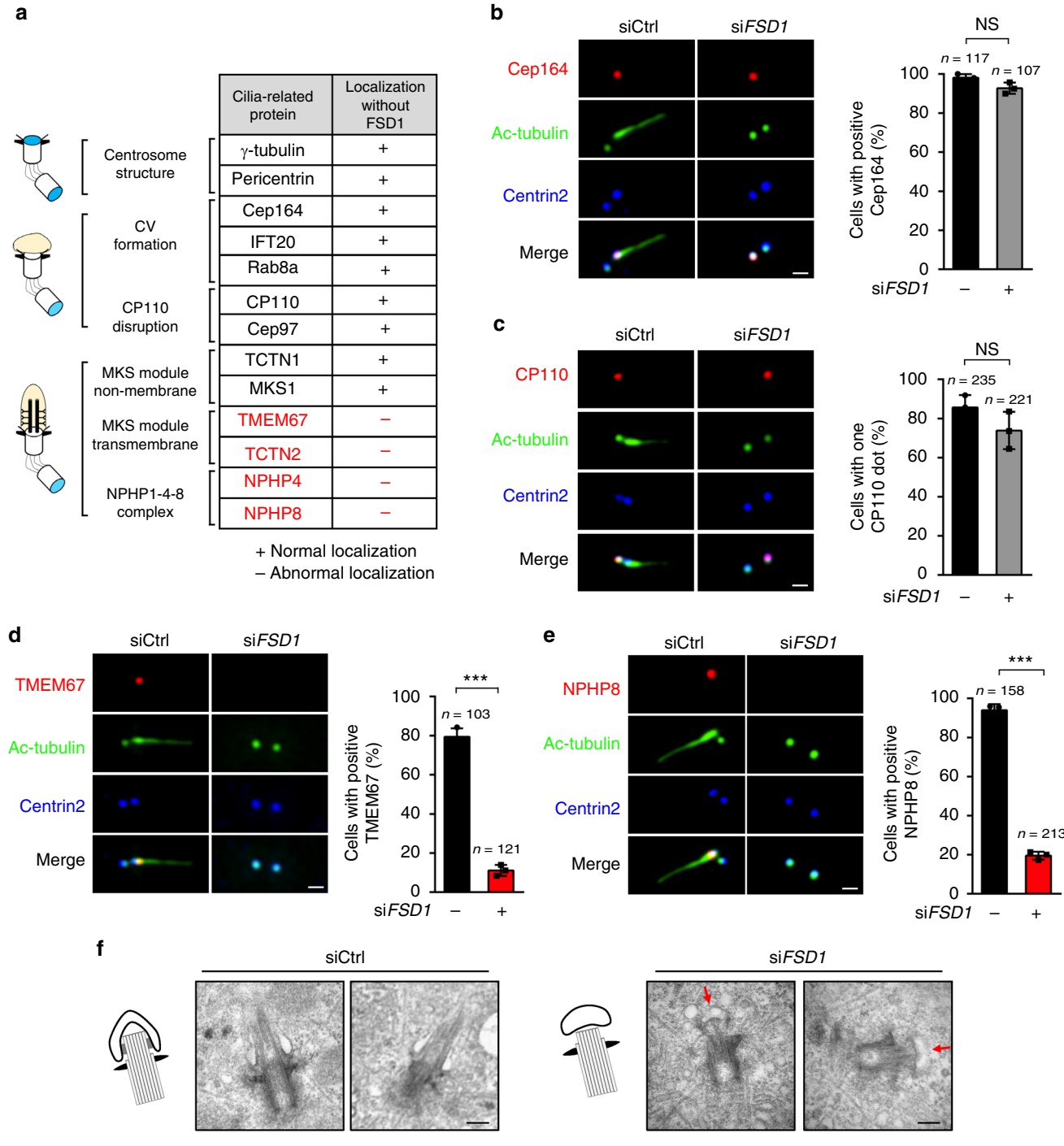

**Fig. 2** Loss of FSD1 blocks ciliogenesis at the stage of transition zone assembly. **a** A table summarizing the centrosome localization of the tested proteins in FSD1-depleted RPE-1 cells. CV ciliary vesicle. **b**–**e** RPE-1 cells were treated with control or FSD1 siRNAs followed by serum starvation for 48 h, then subjected to staining with indicated antibodies. Scale bars, 1 μm. **b** The Cep164 localization was not affected by FSD1 depletion. **c** FSD1 depletion did not affect the disruption of CP110 at the mother centriole. **d**, **e** Loss of FSD1 affected the localization of TMEM67 and NPHP8. **f** Depletion of FSD1 caused ciliary transition zone assembly defects. Representative electron micrographs of basal bodies in RPE-1 cells transfected with control (left) or FSD1 (right) siRNA, followed by serum starvation. Arrows indicate small membrane vesicles connected to FSD1-deficient centrioles through the distal appendage. Schematic diagrams summarizing the phenotype are shown. Scale bars, 200 nm. Data are presented as mean ± s.d. of three independent experiments. *n* number of cells. In all panels, statistical comparisons between two groups were carried out by two-tailed *t*-test. NS not significant, ***$P < 0.001$

localization of Cep290 on CS, but not on the centriole before serum starvation (Fig. 4b). Unexpectedly, FSD1 depletion also decreased the percentage of cells with PCM1 granules around centrosomes (Fig. 4c).

PCM1 is a well-established CS protein that participates in ciliogenesis in a Cep290-dependent manner[38]. To test whether the primary role of FSD1 is the regulation of PCM1, we examined the centriole localization of two well-known downstream proteins of PCM1 in FSD1-depleted cells. Unlike the knockdown of PCM1[20], FSD1 knockdown did not affect the centriolar localization of Pericentrin or Centrin2 (Supplementary Fig. 6d). These results suggest that FSD1 depletion does not completely inactivate PCM1.

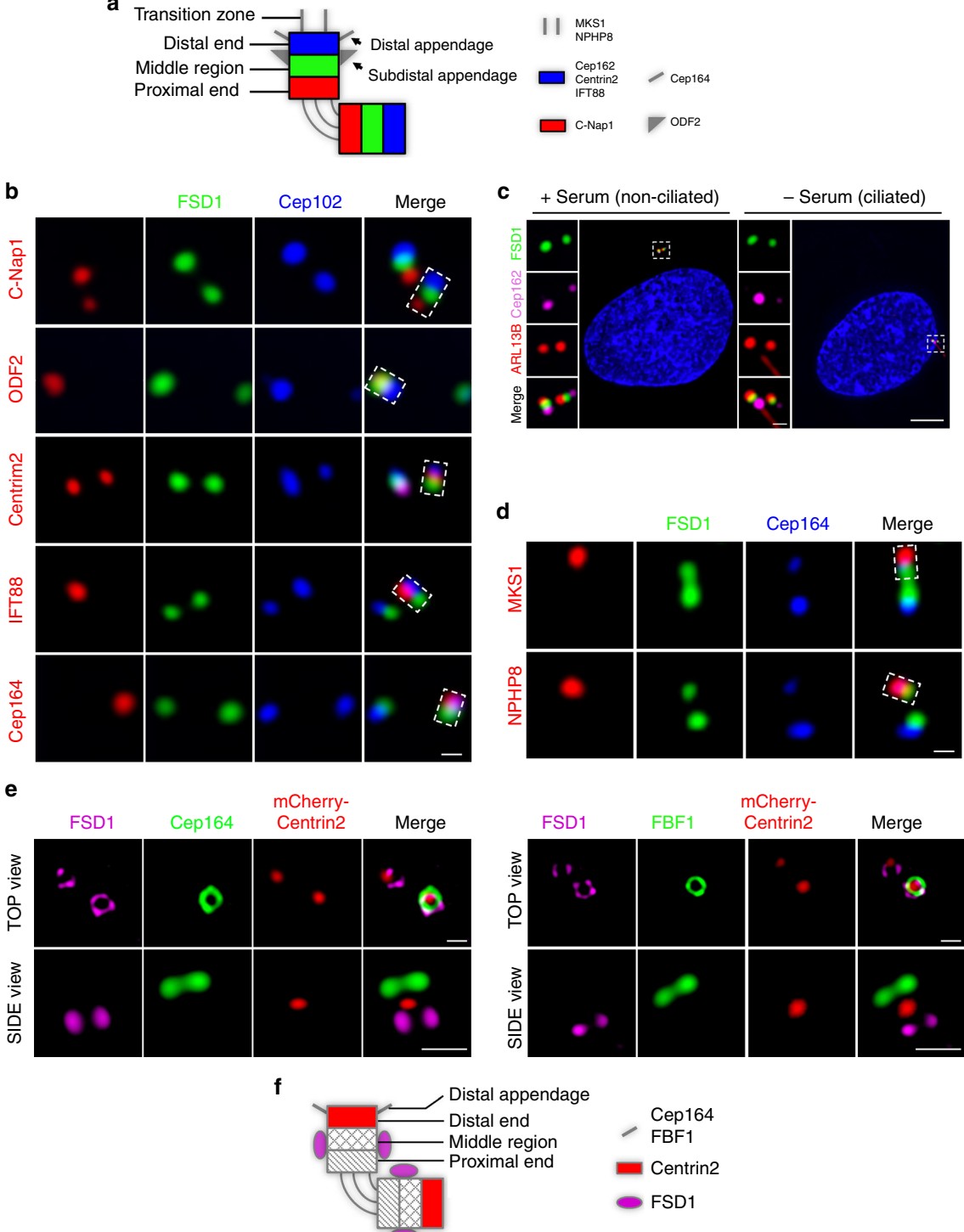

**Fig. 3** FSD1 forms a ring structure at the subdistal region of both centrioles. **a** Schematic of centrosome structure. The centrosome is composed of a pair of centrioles, and each centriole is divided into three portions, including proximal end (red), middle region (green), and distal end (blue). The older centriole of the pair (the mother) is often decorated with distal appendages and subdistal appendages. During ciliogenesis, the transition zone extends from the distal end of the mother centriole. **b** FSD1 was visualized with centriolar proximal (C-Nap1), distal (Cep162, Centrin2, and IFT88), subdistal appendage (ODF2), and distal appendage (Cep164) markers. ODF2, IFT88, and Cep164 localize to mother centrioles. Scale bar, 500 nm. **c** The centriolar localization of FSD1 was observed in ciliated and non-ciliated cells stained with indicated antibodies. Cep162, the centriole distal end marker. ARL13B, the primary cilia marker. Scale bars, 5 μm (main image) and 500 nm (magnified region). **d** The centriolar localization of FSD1 was observed in ciliated cells stained with indicated antibodies. Cep162, the centriole distal end marker. MKS1 and NPHP8, the ciliary transition zone markers. Scale bar, 500 nm. **e** RPE-1 cells were transfected with mCherry-Centrin2 plasmids and stained with the indicated antibodies. The samples were then imaged using the stimulated emission depletion microscopy (STED). Cep164 and FBF1 localize to the distal appendage of mother centrioles. Scale bars, 500 nm. **f** Schematic representation of the centrosome illustrates that FSD1 forms a ring structure at the middle region encircling both centrioles

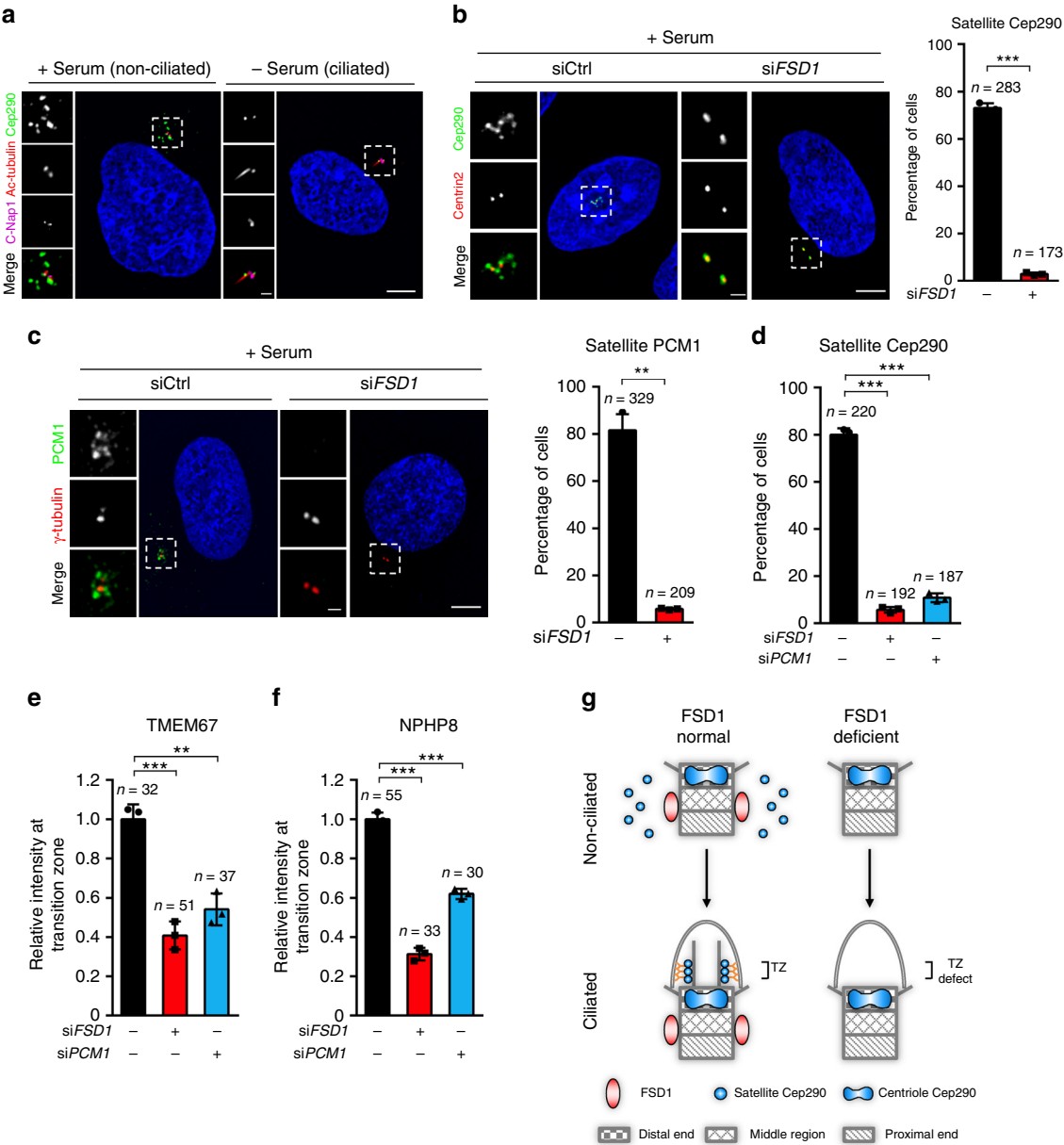

**Fig. 4** The maintenance of Cep290 by FSD1 at CS is required for TZ assembly. **a** Non-ciliated (+Serum) or ciliated (−Serum) RPE-1 cells were stained with indicated antibodies as shown. Scale bars, 5 μm (main image) and 1 μm (magnified region). **b** Effects of FSD1 depletion on the localization of Cep290 at centriolar satellites in cycling cells. RPE-1 cells transfected with control or FSD1 siRNAs were stained with anti-Cep290 (green) and anti-Centrin2 (red) antibodies. Scale bars, 5 μm (main image) and 1 μm (magnified region). **c** Effects of FSD1 depletion on the PCM1 localization at centriolar satellites in cycling RPE-1 cells. RPE-1 cells transfected with control or FSD1 siRNAs were stained with indicated antibodies. Scale bars, 5 μm (main image) and 1 μm (magnified region). **d** Effects of FSD1 or PCM1 depletion on the Cep290 localization at centriolar satellites in cycling RPE-1 cells. **e**, **f** Effects of FSD1 or PCM1 depletion on the relative intensity of the transition zone components TMEM67 (**e**) and NPHP8 (**f**) at mother centrioles in quiescent cells. **g** Model accounting for the TZ assembly defects caused by FSD1 depletion due to the disruption of CS Cep290. TZ transition zone, CS centriolar satellite. Data are presented as mean ± s.d. of three independent experiments. *n* number of cells. In all panels, statistical comparisons between two groups were carried out by two-tailed *t*-test. **P < 0.01, ***P < 0.001

Intriguingly, in FSD1-depleted cells, PCM1 disappears rather than disperses from centriolar satellites (Fig. 4c). A previous study reported that centriolar satellite integrity is very sensitive to cell stress mediated by the p38 pathway[55]. We thus examined PCM1 and Cep290 localization in FSD1-knockdown cells treated with the p38 inhibitor (SB203580) (Supplementary Fig. 6e). The disappearance of PCM1 induced by FSD1 siRNA was indeed prevented by SB203580. Instead, dispersion of PCM1 was observed. Importantly, the disappearance of Cep290 induced by FSD1 knockdown remained unchanged. These results suggest

that the disappearance of PCM1 induced by FSD1 siRNA might potentially be a p38-mediated transfection stress response (Supplementary Fig. 6e). The protein levels of centriolar satellite components PCM1 and Cep290 examined here were not affected by FSD1 depletion (Supplementary Fig. 6f). We hypothesize that PCM1 might exist in another form after FSD1 depletion, so that they could not be easily detected by our immunofluorescence assay, as Centrin2 and Pericentrin, downstream proteins of PCM1, still localized to centrosomes. Thus, our data indicate that FSD1 is essential for the localization of PCM1 and Cep290 at CS.

**Cep290 at CS is required for the TZ assembly**. Cep290, but not PCM1, is a dynamic component of TZ, and rapidly equilibrates with cytoplasmic Cep290 during ciliogenesis[30,38]. We thus wondered whether FSD1 might modulate TZ assembly through disrupting Cep290 at CS and consequently blocking Cep290 trafficking into TZ. We first examined whether the specific ablation of CS Cep290 by PCM1 depletion could also lead to TZ assembly defects, similar to FSD1 depletion. Consistent with previous studies[38], PCM1 depletion in cycling RPE-1 cells led to a marked decrease in the percentage of cells with Cep290 at CS, but did not affect Cep290 at centrioles before cilia formation (Fig. 4d and Supplementary Fig. 6g).

We further found that PCM1 depletion also disrupted the recruitment of TZ proteins, including TMEM67 and NPHP8 (Fig. 4e, f and Supplementary Fig. 6h, i), supporting the possibility that Cep290 at CS might play an essential role in ciliary TZ assembly. Cep290 localizes to both CS and centriolar distal end before cilia formation[38]. During cilia formation, Cep290 disappear from CS[35]. Our data provide a potential possibility that FSD1 depletion causes dramatic decrease of Cep290 localization at CS in interphase, leading to subsequent TZ assembly defects during ciliogenesis (Fig. 4g).

**Localization of CS proteins depends on dynamic MTs**. We next investigated how FSD1 regulated Cep290 or PCM1 localization at CS. A previous report suggested that Cep290 localization at CS requires polymerized MTs, and Cep290 can move along MTs toward the centrosome[38]. Consistent with this report, we observed a substantial decrease in the percentage of cells with Cep290 or PCM1 staining at CS after treatment with the MT-depolymerizing drug, nocodazole (Fig. 5a and Supplementary Fig. 7a, b). We found that treatment with the MT-stabilizing drug Taxol also decreased the percentage of cells with Cep290 or PCM1 staining at CS (Fig. 5b and Supplementary Fig. 7c, d). These results suggest that the CS localization of these two critical CS proteins depends on dynamic MTs.

The transport of many cellular cargos along the MTs depends on the minus end directed motor, dynein[56]. We next investigated the possible involvement of dynein in the localization of Cep290 at CS by using ciliobrevin D, a specific inhibitor of dynein[57]. We found that ciliobrevin D abolished Cep290 staining at CS, but not at centrioles (Fig. 5c and Supplementary Fig. 7e). Consistent with a previous report[58], the PCM1 staining at CS was also affected by high-dose ciliobrevin D addition (Supplementary Fig. 7f). This result suggests that CS proteins, including Cep290 and PCM1, are transported by dynein along MTs.

**FSD1-mediated MT aster formation contributes to TZ assembly**. In interphase, MT nucleation occurs at multiple sites in the centrosome at various stages, and the newly formed MTs are then mainly anchored to the subdistal region of the mother centriole[59]. Because FSD1 is localized to the middle region, which is close to the subdistal region, we examined the potential role of FSD1 in the anchoring of MTs at the centrosome. We observed the regrowth process of MTs after nocodazole release in the FSD1-depleted cells. Five minutes after nocodazole release, both the control and FSD1-depleted cells exhibited strong radial MTs from centrosomes (Fig. 5d). Twenty minutes after nocodazole release, MTs around centrosomes arrayed radially in control cells, but MTs around centrosomes in FSD1-knockdown cells grew in random directions (Fig. 5d and Supplementary Fig. 8a). In contrast to control cells, more than 80% of FSD1-depleted cells had few detectable radial MTs anchored around centrosomes after nocodazole release for 20 min (Fig. 5d, right). Thus, these results

suggest that FSD1 might specifically affect MT anchoring, but not nucleation.

Expectedly, the accumulation of Cep290 at CS was also blocked in most of the FSD1-depleted cells after nocodazole release for 20 min (Fig. 5e). In contrast, depletion of Cep290 at CS and at centrioles did not affect the radial MT regrowth and distribution (Fig. 5e). These findings indicate that MT asters around centrosomes might act upstream of CS Cep290.

It is generally accepted that the newly formed MTs are anchored at the centrosome by many subdistal appendage proteins[14]. We next examined whether the subdistal appendage proteins Ninein and Kif3a, two well-known MT-anchoring proteins, could also regulate the localization of CS proteins and TZ assembly. Although Mazo et al. failed to observe defects in MT anchoring or ciliogenesis in CRISPR/Cas9-mediated Ninein knockout cells (possibly due to compensatory mutations or off-target effects caused by stably expressing Cas9)[19], our following results are consistent with most previous studies focusing on Ninein-mediated MT anchoring[60,61]. Similar to FSD1 depletion, the knockdown of either Ninein or Kif3a, dramatically reduced the localization of Cep290 at CS (Fig. 5f and Supplementary Fig. 8b–d). In contrast, Cep290 still localized normally to the centriole in cells with Ninein or Kif3a depletion (Supplementary Fig. 8b). Expectedly, the MT anchoring at the centrosome and the accumulation of Cep290 at CS after nocodazole release for 20 min was also blocked in most of the Ninein- or Kif3a-depleted cells (Supplementary Fig. 8e). The Ninein- or Kif3a-depleted cells also failed to recruit TMEM67 and NPHP8 to TZ (Fig. 5g, h and Supplementary Fig. 8f, g). Taken together, these results indicate that MT asters anchored at the centrosome by some anchorage proteins, FSD1, Ninein, and Kif3a, are required for CS localization of Cep290 and subsequent ciliary TZ assembly. These results also suggest that FSD1 functions likely through anchoring MT asters at the centrosome.

**MT asters are anchored by FSD1 at mother centrioles**. Next, we investigated how FSD1 affected the anchorage of MT asters at the mother centriole. We found that FSD1 depletion had little effect on the localization of the subdistal appendage protein Ninein (Fig. 6a). In contrast, Kif3a depletion dramatically reduced the Ninein localization on the subdistal appendage as previously reported[18]. Other anchorage proteins, including EB1 and ODF2, localized normally at centrioles in the absence of FSD1 (Supplementary Fig. 9a, b). Our co-localization studies with super-resolution imaging further showed that the FSD1 ring did not completely overlap with the ring formed by ODF2 (Fig. 6b, top view). FSD1 localized at a centriolar middle region, which was very close to and beneath Ninein at the subdistal appendage (Fig. 6b, side view).

We tested the possibility that FSD1 directly anchors MT asters. Indeed, we observed that FSD1 formed a ring at the center of MT asters through super-resolution structured illumination microscopy (SIM) (Fig. 6c). We further found that in addition to its centrosome localization, ectopically expressed full-length FSD1 co-localized with MT asters regrown after nocodazole release for 10 min (Fig. 6d). Consistent with previous reports[41,42], ectopically expressed full-length FSD1 associated with MTs and caused MT bundling (Fig. 6e and Supplementary Fig. 9c). Given that FSD1 contains a coiled-coil domain (CC), a fibronectin type III domain (Fib), and a B30.2/SPRY domain (SPRY), we constructed several truncations for further analysis. The FSD1-SPRY domain (258–496 aa) was capable of MT association, and the FSD1-CC domain (1–172 aa) was sufficient to localize to the centrosome, while the FSD1 ΔCC mutant (154–496 aa) lost its centrosome localization but retained its MT association (Fig. 6e and

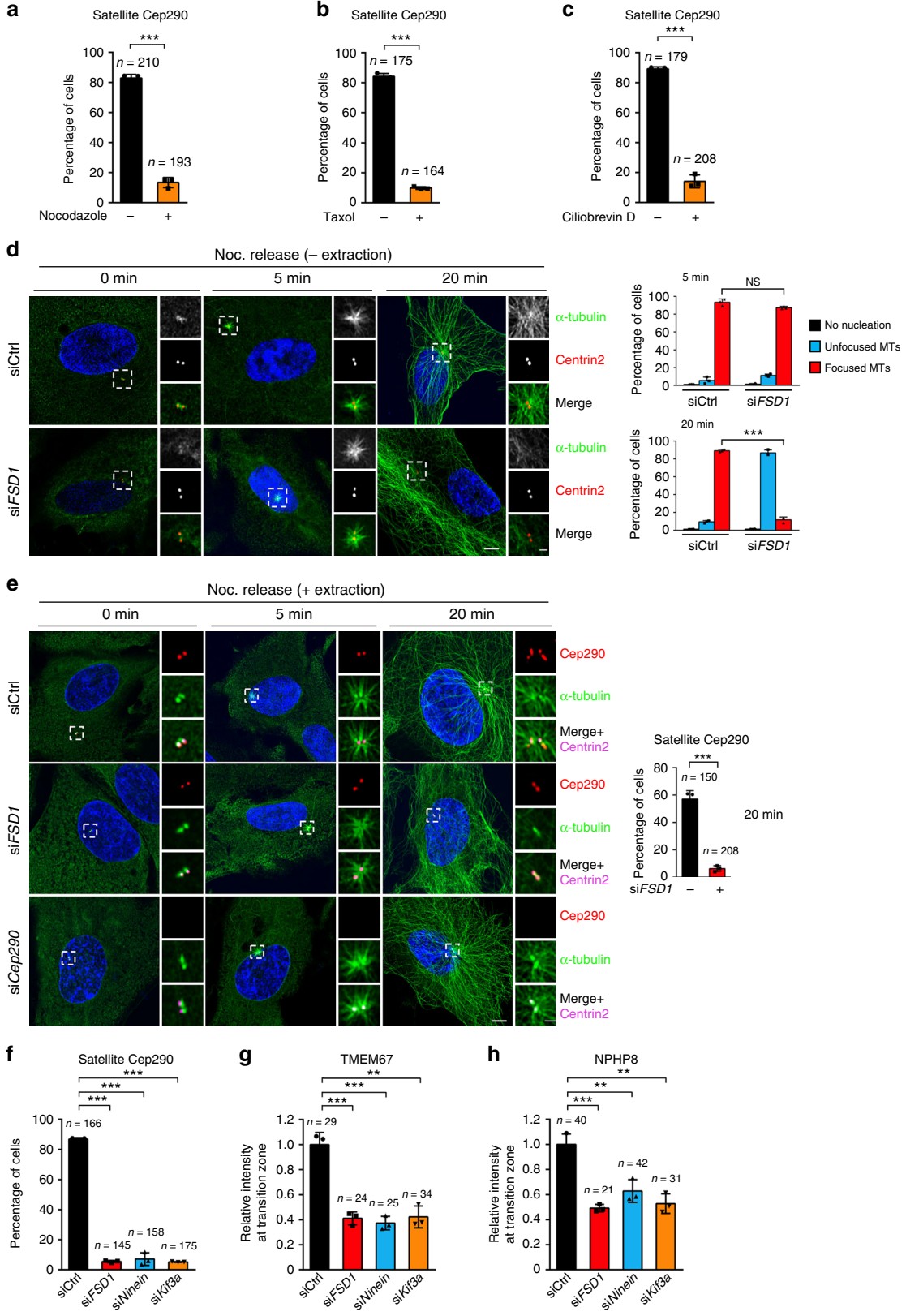

Supplementary Fig. 9c). We also observed that FSD1 ΔCC and SPRY mutants exhibited weak association with centrosomes, which might be due to their binding to centrosome MTs. These data indicate that distinct domains of FSD1 mediate its centrosome localization and MT association.

To determine whether FSD1 binds directly to MTs, we performed a MT-pelleting assay. Endogenous FSD1 from RPE-1 cell lysates and the recombinant FSD1-SPRY domain were co-pelleted with Taxol-stabilized MTs (Fig. 6f, g). In contrast, recombinant FSD1-CC and ΔSPRY (1–268 aa) had no

**Fig. 5** MT aster formation is required for Cep290 localization at CS and TZ assembly. **a–c** Percentage of cells with CS Cep290 after treatment with DMSO, 20 μM nocodazole (**a**), 1 μM taxol (**b**), or 10 μM ciliobrevin D (**c**) for 30 min. **d** RPE-1 cells transfected with control or FSD1 siRNA were subjected to nocodazole treatment for 2 h and released at indicated time points. Next, cells were fixed with 4% paraformaldehyde in PBS and stained with antibodies to α-tubulin to visualize microtubules (green) and Centrin2 to mark centrosomes (red). Percentage of cells with different microtubule regrowth statuses at indicated time points was quantified on the right panel. Scale bars, 5 μm (main image) and 1 μm (magnified region). **e** RPE-1 cells were transfected with indicated siRNA were subjected to nocodazole treatment for 2 h and released at indicated time points. Cells were then fixed and permeabilized with cold methanol and stained with antibodies to Cep290 (red), α-tubulin (green), and Centrin2 (purple). Percentage of cells with CS Cep290 after nocodazole release for 20 min was quantified on the right panel. Scale bars, 5 μm (main image) and 1 μm (magnified region). **f** Effects of FSD1, Ninein, or Kif3a depletion on the Cep290 localization at centriolar satellites in cycling RPE-1 cells. **g, h** Effects of FSD1, Ninein, or Kif3a depletion on the relative intensity of the transition zone components TMEM67 (**g**) and NPHP8 (**h**) at mother centrioles in quiescent cells. Data are presented as mean ± s.d. of three independent experiments. *n* number of cells. In all panels, statistical comparisons between two groups were carried out by two-tailed *t*-test. NS not significant, **P < 0.01, ***P < 0.001

MT-binding activity (Fig. 6g) indicating that FSD1 directly binds to MTs through its SPRY domain. We further found that GFP-FSD1 could bind to MT specifically in vitro as revealed by total internal reflection microscopy (Fig. 6h and Supplementary Fig. 9d). Interestingly, a small portion of GFP-FSD1 could directly bind to the MT minus ends. Taken together, our results strongly suggest that the dual activities of FSD1 in both centrosome and MT binding allow it to anchor MT asters at centrosomes.

**The CAP350-FOP complex mediates the localization of FSD1.** Next, we tried to identify possible proteins that regulate the FSD1 localization at centrosomes by knocking down well-established MT-anchoring proteins[21,24,26,27]. Among 12 MT-anchoring proteins tested, we found that only two MT-anchoring proteins, CAP350 and FOP, are required for the localization of FSD1 at centrosomes (Fig. 7a and Supplementary Fig. 9e). The CEP19/CAP350/FOP module, localized as rings at the distal end of centrioles, are required for ciliogenesis and for anchoring MTs at centrosomes at least in part through Ninein[21,22]. FOP and CAP350 are mutually dependent for the localization of each other, and both act upstream of CEP19[21,22]. Our data further showed that FSD1 is also required for the localization of CAP350, but not of FOP (Fig. 7b). CEP19 is dispensable for the localization of FSD1 and vice versa (Fig. 7b).

We further found that both CAP350 and FOP were able to bind to ectopically expressing Flag-FSD1, whereas CEP19 did not show any detectable association with Flag-FSD1 (Fig. 7c). We then examined whether CAP350-FOP and FSD1 interacted in more physiological conditions by immunoprecipitating endogenous proteins. Both CAP350 and FOP were present in anti-FSD1 immunoprecipitates, and vice versa (Fig. 7d). Neither CAP350-FOP nor FSD1 coprecipitated when we used rabbit immunoglobulin G as the control antibody. Taken together, our data reveal that FSD1 is likely recruited to centrioles by the CAP350-FOP complex, and FSD1 contributes to the localization of CAP350 at the centrosome in a positive feedback (Fig. 7e).

**The MT-binding activity of FSD1 is critical in ciliogenesis.** Many MT-associated proteins are characterized by a region of basic amino acids, which binds to the acidic C-terminal tails of tubulin[62]. We made an alignment of FSD1-SPRY domain and identified six highly conserved arginines in this domain (Fig. 8a). We generated a series of FSD1 mutants, in which these arginines were substituted with glutamic acid. Unlike wild-type (WT) FSD1 and other mutants, the FSD1 3RE mutant (R332E, R335E, and R337E) failed to induce MT bundles in the cytoplasm and radial MT formation around centrosomes (Fig. 8b). Importantly, this mutant retained localization at centrosomes (Fig. 8b). In addition, the mutation of these three arginines also eliminated the MT-

binding activity of FSD1 in vitro (Fig. 8c). More importantly, the expression of siRNA-resistant WT FSD1, but not the 3RE mutant, could efficiently rescue the defects in MT aster formation induced by FSD1 knockdown (Fig. 8d). Thus, these results suggest that FSD1 promotes the anchoring of MT asters at centrosomes by directly binding MTs.

We further analyzed the effects of WT or 3RE mutant on ciliogenesis in FSD1-knockdown cells. We found that WT FSD1 could rescue the defects in ciliogenesis, Cep290 and PCM1 localization at CS, and TZ assembly caused by FSD1 depletion (Fig. 8e–h and Supplementary Fig. 9g). In contrast, the 3RE mutant without MT-binding activity could not rescue these defects induced by depletion of FSD1, although it is still localized to centrosomes (Fig. 8e–h and Supplementary Fig. 9g). The levels of ectopically expressed FSD1 WT or 3RE mutant proteins were comparable to that of the endogenous FSD1 (Supplementary Fig. 9f). Thus, these data indicate that the critical role of FSD1 in Cep290 localization at CS and TZ assembly depends on its MT-binding activity.

## Discussion

Our results presented in this study indicate that FSD1 facilitates MT aster formation by anchoring MTs at mother centrioles, which in turn maintains the dynamics of CS in interphase. This process likely enables satellite Cep290 to subsequently promote TZ assembly during ciliogenesis. In cells lacking FSD1, MTs become disorganized, and Cep290 fails to localize to the CS, resulting in TZ assembly defects (Fig. 8i).

MTs nucleated at or near the centrosome can be readily anchored at the subdistal region of the mother centriole[60]. Our data now reveal a critical role of FSD1 in anchoring MT asters at the centrosome by directly binding MTs. FSD1 forms a ring structure at the middle region of both mother and daughter centrioles and directly binds to MTs through its SPRY domain. As a core subdistal appendage protein, Ninein is well known to nucleate and anchor MTs to mother centrioles. Depletion of FSD1 did not affect the localization of Ninein and vice versa, suggesting that they are independently recruited to the centrioles. Both CAP350 and FOP are required for the localization of FSD1 on centrioles and physically interact with FSD1. These proteins serve as the centrosome receptor of FSD1.

Unlike the subdistal appendage protein Ninein, FSD1 localizes to the contiguous zone beneath the subdistal appendage, suggesting that this narrow zone also acts as an anchorage site for MT asters. Based on our data and a previous report[60], both subdistal appendages and this contiguous zone of the mother centriole appear to be MT-anchoring structures that promote MT aster formation during interphase. Although FSD1 also localizes to the daughter centriole, FSD1 at the daughter centriole is not sufficient to anchor MT asters. Thus, FSD1 possibly collaborates

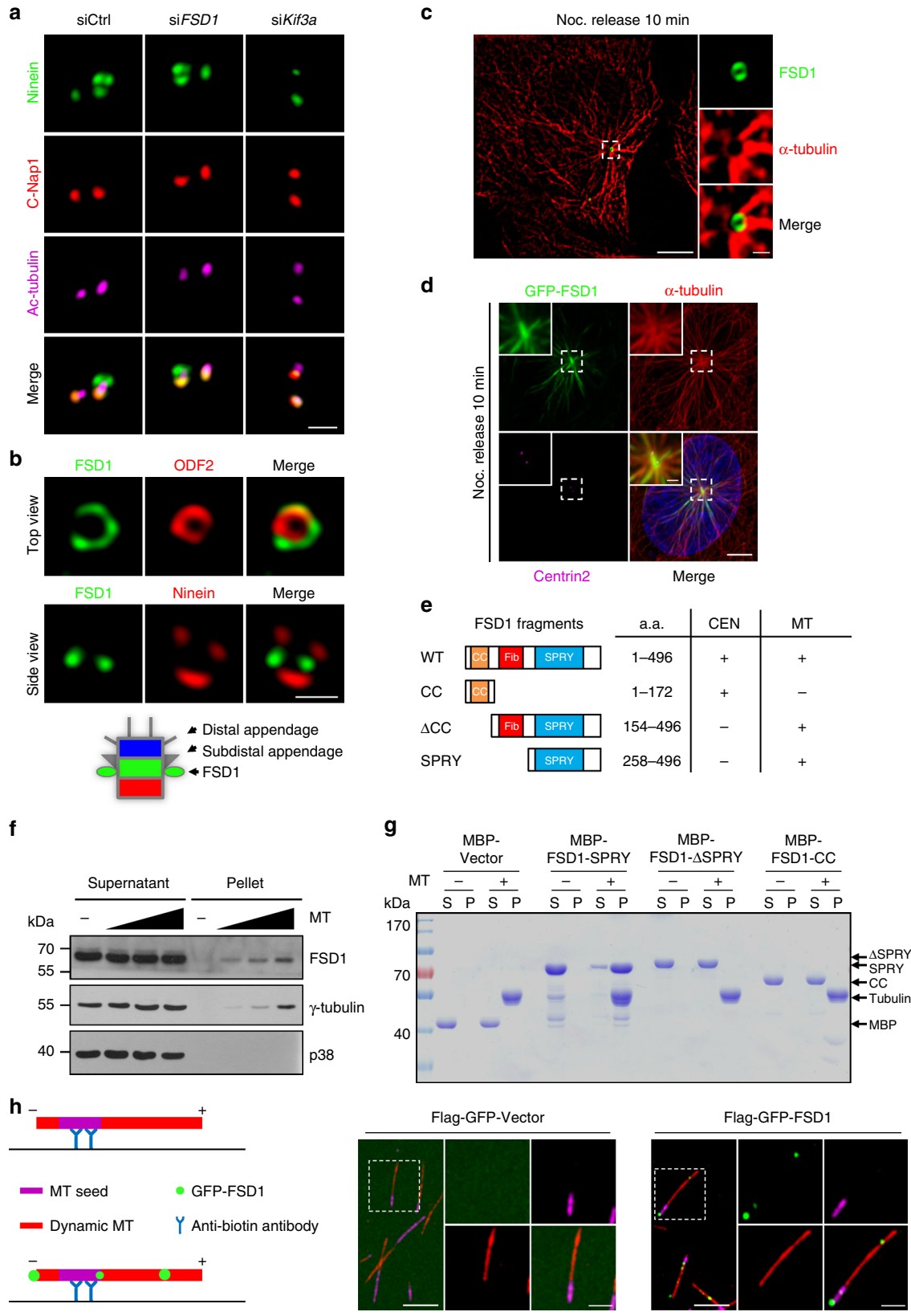

with other proteins at the middle region of the mother centriole to form higher affinity binding sites for MT asters.

The 3RE mutant of FSD1, which cannot anchor MTs to centrosomes but retain localization at centrosomes, fails to rescue CS Cep290 delocalization and ciliary TZ assembly defects in FSD1-knockdown cells. These data indicate that the critical role of FSD1 in ciliogenesis depends on its MT-binding activity, although other possible mechanisms cannot be completely ruled out.

Cep290 has multiple critical functions during ciliogenesis, including CV formation and TZ assembly[63]. Before cilia

**Fig. 6** MT asters are anchored by FSD1 at mother centrioles. **a** RPE-1 cells transfected with control, FSD1, or Kif3a siRNAs were stained with anti-Ninein (green), anti-C-Nap1 (red), and anti-Ac-tubulin (purple) antibodies. Scale bar, 1 μm. **b** RPE-1 cells stained with indicated antibody were visualized using the stimulated emission depletion microscopy (STED). Schematic representation of the precise FSD1 localization on mother centrioles. Scale bar, 500 nm. **c** RPE-1 cells subjected to a microtubule regrowth assay were fixed and permeabilized with cold methanol and stained with antibodies against FSD1 and α-tubulin. The sample was then imaged through super-resolution 3D-SIM. Scale bars, 5 μm (main image) and 500 nm (magnified region). **d** Ectopically expressed full-length FSD1 co-localized with microtubule asters regrown after nocodazole release 10 min. Magnified centrioles are shown in the insets. Scale bars, 5 μm (main image) and 1 μm (magnified region). **e** Schematic diagram illustrates the different domains of FSD1 for its microtubule (MT) binding and centriolar (CEN) localization. **f** Endogenous FSD1 binds to microtubules. RPE-1 cell extracts were incubated with increasing amounts of Taxol-stabilized microtubules and then centrifuged (100,000 × g for 40 min) to form supernatant and pellet fractions. Samples from both fractions were probed with the indicated antibodies. The p38 and γ-tubulin were used as negative and positive controls for microtubule binding, respectively. **g** The FSD1-SPRY domain could directly bind to microtubules. The different purified MBP-tagged FSD1 truncations were incubated in the presence (+) or absence (−) of Taxol-stabilized microtubules (MTs) and were separated into supernatant (S) and pellet (P) fractions after high-speed centrifugation. Coomassie blue staining of both tubulins and FSD1 is indicated. **h** FSD1 could bind to microtubules (MTs), as shown by TIRF microscopy. GMPCPP seed MTs (with Alexa-647 and biotin labeled tubulin) were attached to a neutravidin-coated coverslip. Unpolymerized tubulin labeled with Alexa 561 was added and allowed to polymerize at room temperature for 5 min, then, 10 nM Flag-GFP-Vector or Flag-GFP-FSD1 (green) was added to the solution. The sample was imaged after reaction setup for 5 min through TIRF microscopy. Scale bars, 5 μm (main image) and 2 μm (magnified region)

formation, Cep290 localizes to both CS and centrioles[38,64]. During cilia formation, Cep290 no longer localizes to CS and only localizes to centrioles[35]. In *Chlamydomonas* and mammalian cells, Cep290, but not other CS proteins, has been shown to localize to the TZ during ciliogenesis[30,32]. FSD1-depleted cells exhibit normal CV formation, but defective TZ assembly. In addition, inactivation of FSD1 specifically causes the loss of Cep290 at CS, but not centrioles, suggesting a critical role of CS Cep290 in TZ assembly. Depletion of satellite PCM1 or other MT-anchoring proteins (Ninein or Kif3a) also specifically eliminates Cep290 from CS and causes defective TZ assembly, further supporting this notion.

At present, we do not know how Cep290 at CS promotes TZ assembly. Our findings suggest that the CS pool of Cep290 needs to be actively maintained by dynein-dependent transport along MT asters, and this process requires FSD1. Whether this process is corroborated by other CS proteins remains to be addressed.

During ciliogenesis, doublet MTs vertically extending from mother centrioles form a structural backbone for cilia. Our data suggest that the horizontal MT asters anchored at mother centrioles might also be required for the assembly of vertical ciliary axoneme. Perturbation of MT aster dynamics or inhibition of MT-dependent motor dynein disrupts the localization of Cep290 or PCM1 at CS. Our results thus suggest that MT asters act upstream of CS proteins and contribute to their dynamic maintenance.

Our results establish a critical role of MT asters anchored at the centrosome in maintaining Cep290 at CS in interphase cells and TZ assembly during ciliogenesis. Our findings uncover an important molecular pathway for ciliogenesis and reveal molecular targets for the diagnosis and potential treatment of ciliopathies. By virtue of its ability to link canonical MTs to doublet MTs in cilia, the centrosome serves as a versatile signaling platform to enable the effective communication between the cell and extracellular environment.

## Methods

**Cell culture**. U2OS cells were obtained from the American Type Culture Collection and cultured in Dulbecco's modified Eagle's medium (DMEM) medium with 10% fetal bovine serum (FBS) and 1% penicillin/streptomycin. Human hTERT-RPE-1 (RPE-1) cells were kindly provided by Xueliang Zhu and cultured in DMEM/F-12 (1:1) medium supplemented with 10% FBS, 0.01 mg/ml hygromycin B, and 1% penicillin/streptomycin. For cilia formation, RPE-1 cells were starved in Opti-MEM reduced serum media (Life Technologies) for 48 h, except the examining the TZ structure assay in Fig. 2f.

**Cloning and plasmids**. The plasmid expressing recombinant GFP-FSD1 was obtained from Alan Rick Horwitz. The GFP-FSD1 siRNA-resistant plasmids and

GFP-FSD1-3RE mutant were generated by PCR-based site-directed mutagenesis. The GFP-FSD1 truncation plasmids were amplified by PCR and cloned into pEGFP-N1. The Flag-FSD1, Flag-GFP-Vector, and Flag-GFP-FSD1 plasmids were amplified by PCR and cloned into pcDNA3.0. The MBP-FSD1-WT, SPRY, ΔSPRY, and CC plasmids were amplified by PCR and cloned into pOCC22-pET11 to generate MBP-tagged recombinant proteins, respectively. The pOCC22-pET11 plasmid was obtained from Xin Liang. All the primers used for cloning and real-time PCR were listed in Supplementary Table 3, 4. All constructs were verified by DNA sequencing. Plasmid transfection into RPE-1 cells was performed using Lipofectamine™ LTX Reagent (Life Technologies) according to the manufacturer's instruction.

**RNAi**. Synthetic siRNA oligonucleotides were obtained from Life Technologies. Transfection of siRNAs using RNAiMAX (Life Technologies) was performed according to the manufacturer's instructions. The sequences of siRNAs (Invitrogen) are as follows.

Control siRNA: 5′-UUCUCCGAACGUGUCACGUAA-3′.
*FSD1* siRNA (1#): 5′-GCAGGAUAUCAAGGCUCGCGAGAAA-3′.
*FSD1* siRNA (2#): 5′-CCAAGGCCUCCUGUCCUUCUACAAU-3′.
*FSD1* siRNA (3#): 5′-CCGCUGCUGCCUGCUUUCACGGUAU-3′.
*PCM1* siRNA: 5′-AGAUAAUGAAGCUCCACUGACUCCC-3′.
*Ninein* siRNA: 5′-GGAGGCGGAGCUCUCUGAAGUUAAA-3′.
*Kif3a* siRNA: 5′-GCGAUAAUGUGAAGGUUGUUGUUAG-3′.
*FOP* siRNA: 5′-AUAAUAAGGGUCCACCCACAGUACC-3′.
*CAP350* siRNA: 5′-UAAACGUCGCUUCCACUCUAGGAGC-3′.
*ODF2* siRNA: 5′-AUACUCAGCAGUCUGACGGUGUAGU-3′.
*p150^Glued* siRNA: 5′-AACAGCACCAGAACGCAGUCAUGGU-3′.
*EB1* siRNA: 5′-UCAACAGUAAGUUUCAAUACGUUGA-3′.
*EB3* siRNA: 5′-AAAGCUGCUUGCAGCACCUUGAAGU-3′.
*SSX2IP* siRNA: 5′-AGCACUUGCUGUGAGUAUAAACUUG-3′.
*CEP19* siRNA: 5′-GCCUCCAGCUAUUAUCUUA-3′.
*TCHP* siRNA: 5′-CAGGCAGAAUGGAGCUCUA-3′.

**Immunoprecipitation and immunoblotting**. For Flag-tagged proteins' immunoprecipitation experiments, Flag-tagged constructs were transfected into 293T cells, and then cells were harvested 24 h after transfection. Flag M2 beads (Sigma, A2220) were incubated with cell extract at 4 °C for 2 h. For other immunoprecipitation experiments RPE-1 cells were lysed at 4 °C for 30 min with ELB buffer: 50 mM HEPES (pH 7.0), 250 mM NaCl, 5 mM EDTA (pH 8.0), 0.1% NP-40, 10% glycerol, 1 mM dithiothreitol (DTT), 0.5 mM phenylmethylsulfonyl fluoride (PMSF), and phosSTOP. Extracts were cleared by subsequent centrifugations. Immunoprecipitation was carried out with an indicated antibody at 4 °C overnight. Proteins were separated by SDS–polyacrylamide gel electrophoresis and analyzed by western blotting with the indicated antibodies. Images have been cropped for presentation (uncropped scans of the blots were shown in Supplementary Fig. 10). The same results were obtained in at least three separate experiments for each interaction assay.

**Immunofluorescence microscopy**. To visualize centrosome proteins in RPE-1 cells, cells were fixed and permeabilized in −20 °C methanol for 5–10 min. To detect Ninein in RPE-1 cells, before fixation with methanol at −20 °C for 10 min, cells were extracted in PTEM (20 mM PIPES [pH 6.8], 0.2% Triton X-100, 10 mM EGTA, and 1 mM MgCl₂) for 2 min. Fixed cells were blocked with 3% normal goat serum (NGS) in 0.1% Triton X-100/phosphate-buffered saline (PBS) prior to incubation with primary antibodies. Secondary antibodies used were Alexa Fluor 488-, 546-, 594-, 647-, and 405-conjugated donkey anti-mouse, anti-rabbit, or anti-

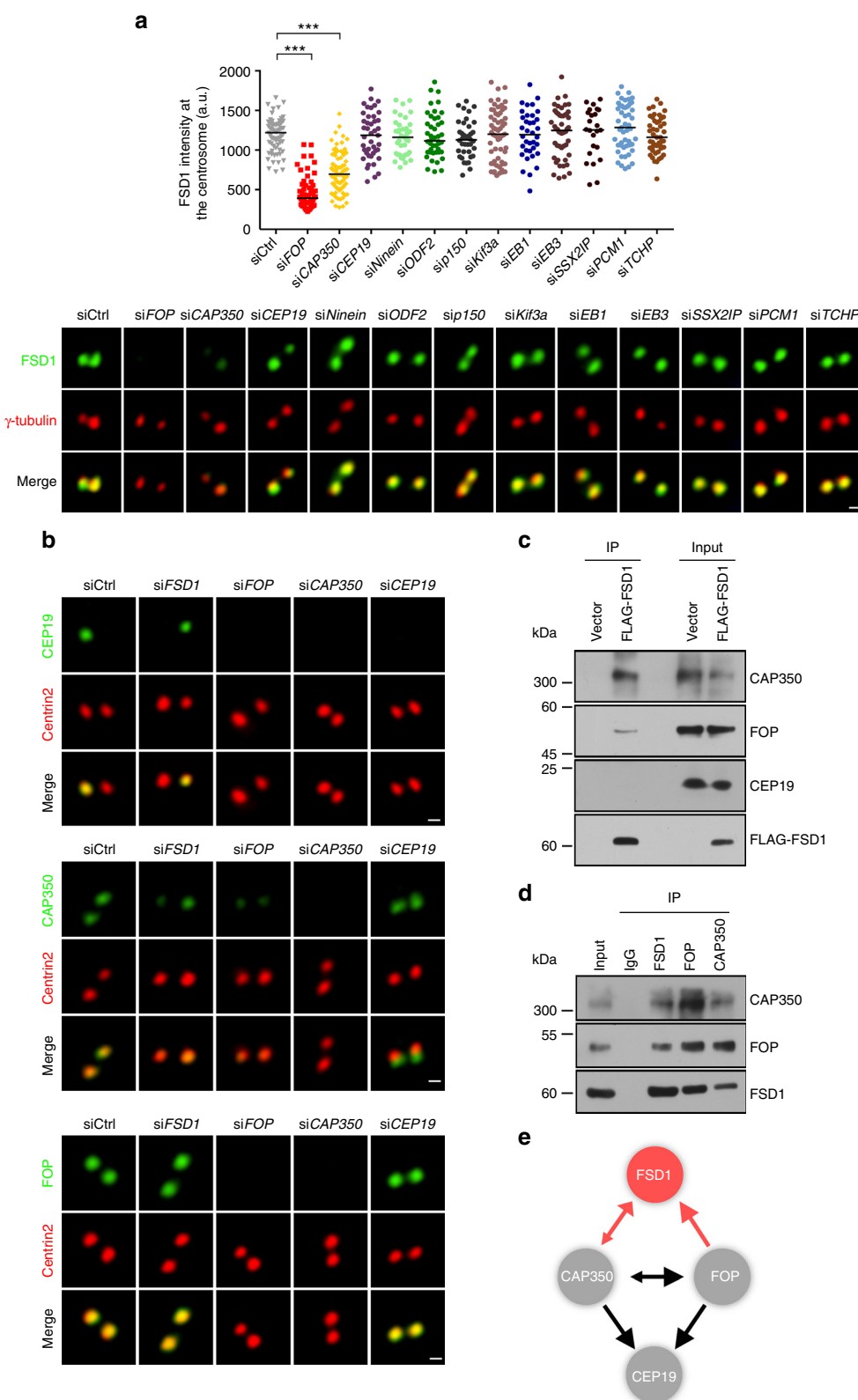

rat IgG (Life Technologies). When two different primary antibodies from the same species were used together, one of the antibodies was directly coupled with a fluorophore by using APEX™ Alexa Fluor™ 488 Antibody Labeling Kit (Life Technologies). After the standard indirect immunofluorescence labeling procedure, cells were incubated with the labeled primary antibody for 1 h at room temperature. DNA was stained with Hoechst 33342 (1:500, H3570, Invitrogen).

Images were acquired with a ×63/1.40 oil objective on a DeltaVision Image Restoration Microscope (Figs. 1, 2, 3, and 7), a ×63/1.40 oil objective on Zeiss LSM 880 (Figs. 4, 6, and 8) or a ×100/1.40 oil objective on Zeiss LSM 880 with airyscan (Figs. 5 and 6) or a ×63/1.40 oil objective on Nikon Eclipse Ti-E Microscope with an Ultra View spinning-disc confocal scanner unit (Perkin Elmer) (Fig. 5). All acquisition settings were kept constant for experimental and control groups in the same experiment. The representative images acquired by DV system were processed by iterative constrained deconvolution (SoftWoRx, Applied Precision Instruments).

All quantification was performed on raw images without deconvolution. Briefly, the raw images were analyzed with Volocity 6.0 software. To quantify the relative

**Fig. 7** CAP350-FOP complex are essential for FSD1 centrosome localization. **a** Effects of depletion of known 12 anchoring proteins on the FSD1 localization at centrosomes. RPE-1 cells were transfected with indicated siRNAs and stained with antibodies to FSD1 (green) and γ-tubulin (red). Bars indicate the median. Significance between two groups was determined by two-tailed t-test. ***$P < 0.001$. Scale bar, 500 nm. **b** Immunofluorescence analysis of the relationship between FSD1 and the CAP350-FOP complex. RPE-1 cells transfected with the indicated siRNAs were stained with antibodies to CEP19, CAP350, or FOP (green), and to Centrin2 (red). Scale bar, 500 nm. **c** Immunoprecipitation and immunoblot analysis of the interaction of Flag-FSD1 with endogenous FOP, CAP350, and CEP19 in HEK293T cells. HEK293T cells were transfected with Flag-tagged FSD1; lysates immunoprecipitated with anti-Flag, as well as input lysates, were analyzed by immunoblot with indicated antibodies. **d** The interaction between endogenous FSD1 and CAP350, FOP in RPE-1 cells. Immunoblot analysis of lysates of RPE-1 cells immunoprecipitated with anti-FSD1, anti-CAP350, anti-FOP, or control IgG. **e** Model depicting the interactions between FSD1 and the CAP350-FOP complex. The size of arrowhead represents the affected strength on corresponding centrosome localization

intensity of centriolar components, centriolar regions were selected based on Centrin2 or Ac-tubulin signal, and an area with the same size was selected in non-centriolar regions for background determination. The relative centriolar intensity of a given protein was calculated as the mean of its background-subtracted centriolar intensity divided by the background-subtracted intensity of centriolar Centrin2 or Ac-tubulin.

Super-resolution imaging was performed on a Leica TCS SP8 STED 3X system with an HCX PL APO ×63 numerical aperture (NA) 1.40 oil objective. Pixel size was <20 nm (typically 15 nm). Images were deconvolved using Huygens Professional software (Scientific Volume Imaging B.V). The 3D-SIM images were taken using electron multiplying charge-coupled device (EMCCD) camera (Andor iXon3 DU-897E) on an N-SIM system (Nikon) equipped with an Apo total internal reflection fluorescence ×100/1.49 oil objective.

**Antibodies**. Antibodies used in this study included mouse anti-β-actin (1:1000, sc-47778, Santa), rabbit anti-Cep97 (1:100, sc-100028, Santa), mouse anti-C-Nap1 (1:100, sc-390540, Santa), rabbit anti-p38 (1:1000, sc-535, Santa), rabbit anti-ARL13B (1:100, 17711-1-AP, Proteintech), rabbit anti-CEP19 (1:100, 26036-1-AP, Proteintech), rabbit anti-C-Nap1 (1:100, 14498-1-AP, Proteintech), rabbit anti-CP110 (1:800, 12780-1-AP, Proteintech), rabbit anti-IFT20 (1:100, 13615-1-AP, Proteintech), rabbit anti-IFT88 (1:100, 13967-1-AP, Proteintech), rabbit anti-MKS1 (1:100, 16206-1-AP, Proteintech), rabbit anti-NPHP4 (1:100, 13812-1-AP, Proteintech), rabbit anti-TMEM67 (1:100, 13975-1-AP, Proteintech), rabbit anti-TCTN1 (1:100, 15004-1-AP, Proteintech), rabbit anti-TCTN2 (1:100, 17053-1-AP, Proteintech), rabbit anti-EB1 (1:200, 17717-1-AP, Proteintech), rabbit anti-EB3 (1:1000, 23974-1-AP, Proteintech), rabbit anti-ODF2 (1:500, 12058-1-AP, Proteintech), rabbit anti-kif3a (1:100, 13930-1-AP, Proteintech), rabbit anti-FOP (1:100, 11343-1-AP, Proteintech), mouse anti-FLAG (1:1000, F3165, Sigma), rabbit anti-FBF1 (1:200, HPA023677, Sigma), mouse anti-γ-tubulin (1:400, T6557, Sigma), rabbit anti-γ-tubulin (1:200, T3559, Sigma), mouse anti-Ac-tubulin (1:400, T6793, Sigma), rabbit anti-RPGRIP1L (1:100, HPA039405, Sigma), α-Tubulin–FITC (1:100, F2168, Sigma), Ki-67-FITC (1:100, 11-5699-82, eBioscience), mouse anti-p150$^{Glued}$ (1:100, 610473, BD Biosciences), mouse anti-Rab8a (1:100, 610844, BD Biosciences), mouse anti-Centrin2 (1:400, 04-1624, Milliopore), rabbit anti-CEP290 (1:200, A301-659A, Bethyl), rabbit anti-Centrin2 (1:200, ab196554, abcam), rabbit anti-Pericentrin (1:400, ab4448, abcam), rabbit anti-Ninein (1:100, A301-504A, Bethyl), rabbit anti-α-tubulin (1:100, #2144, CST), rabbit anti-α-tubulin (1:100, PM054, MBL), rabbit anti-PCM1 (1:500, #5213, CST), rabbit anti-CEP164 (1:600, 45330002, NOVUS), rabbit anti-CAP350 (1:100, NB100-59811, NOVUS), rabbit anti-TCHP (1:100, PAB23320, Abnova), and rat anti-CEP162 (1:100, gift from Meng-Fu Bryan Tsou). To generate monoclonal mouse or poly-clonal rabbit anti-FSD1 antibodies, a glutathione-S-transferase fusion protein containing full-length FSD1 was expressed in *Escherichia coli* and purified to homogeneity as the antigen and used at a 1:500 dilution.

**Zebrafish**. WT AB zebrafish embryos were obtained by natural spawning. All of the adult fishes were maintained in normal system water at 28.5 °C. Antisense MOs were obtained from GeneTools. The target sequences of MOs and gRNA are as follows.

*Fsd1*-aMO: 5′-ACTCCTTCTGGTCGTCCATGCTGTC-3′.
*Fsd1*-sMO: 5′-TGGCCTCCTGTAGTGTTTACCTCTA-3′.
*Fsd1*-misMO: 5′-ACTACTTATGGTAGTCCATACTATC-3′.
*Fsd1* gRNA: 5′-GGGCGCCTGCACCAAAGCCC-3′.

They were injected into blastomere/yolk boundary at one-cell stage. For typical CRISPR experiments, 600 pg of Cas9 mRNA and 25 pg of *fsd1* gRNA were co-injected into zebrafish embryos at one-cell stage. Zebrafish fsd1-aMO-resitant full-length mRNAs or CC mRNA were transcribed from *Not*I-digested pCS2-fsd1 or pCS2-CC plasmid using the mMessage mMachine SP6 kit (Ambion) and co-injected with fsd1-aMOs into zebrafish embryos at one-cell stage at the concentration of 60 pg per embryo. The primers used for *fsd1* mutant genotyping and *fsd1* mRNA synthesis were listed in Supplementary Table 1, 2. For whole-mount in situ hybridization, embryos were fixed in 4% paraformaldehyde and hybridized with Dig-labeled *spaw* or *cmlc* probes at 65 °C. This study was approved

by the Ethical Review Committee of Institute of Zoology, Chinese Academy of Sciences, China.

**Transmission electron microscopy**. RPE-1 cells grown on 35 mm glass bottom dishes (MatTek Corporation) were co-transfected with a plasmid expressing GFP-Centrin2 and indicated siRNAs for 48 h. The culture media was changed to Opti-MEM reduced serum media (Life Technologies) to promote the formation of primary cilia for 24 h. We then performed correlative light and electron micro-scopy strategy to observe the structure of TZ. Briefly, we first selected centrosomes by light microscope and then the same centrosome was imaged on the electron microscope. Cells were fixed for 1 h at room temperature with 4% paraformalde-hyde and 2% glutaraldehyde buffered in PBS buffer. Samples were imaged with ×63/1.40 oil objective on Zeiss LSM 880 system. After target cells were selected and marked on the light microscope, cells were incubated in 0.15% Tannic acid in PBS buffer for 1 min, post-fixed in 2% OsO$_4$ in sodium cacodylate buffer for 1 h on ice, prestained with 1% uranyl acetate diluted in ddH$_2$O for 1 h at 4 °C, dehydrated in a graded series of ethanol, infiltrated with EPON812 resin (Electron Microscopy Sciences), and then embedded in the resin. Serial sections (60–80 nm thickness) were cut on a microtome (Ultracut UC6; Leica) and stained with 1% uranyl acetate as well as 1% lead citrate. Samples were examined on a JOEL transmission electron microscope.

**MT-binding assay**. Purified tubulin proteins (T240) were purchased from Cytoskeleton and diluted to a final concentration of 5 mg/ml in general tubulin buffer (80 mM PIPES, pH 7.2, 0.2 mM MgCl$_2$, and 0.5 mM EGTA including 1 mM GTP). Add 2 μl of Cushion Buffer (60% glycerol and 20 μM Taxol in general tubulin buffer) to one 20 μl aliquot of tubulin Protein and incubate at 35 °C for exactly 20 min. This step allows tubulin to polymerize to MTs, then add 2 μl of 2 mM Taxol stock solution. Taxol-stabilized MTs were mixed with affinity-purified FSD1 proteins from *E. coli*, or with cell extracts (1.5 mg protein) prepared from RPE-1 cells in general tubulin buffer including protease inhibi-tors. After 30 min of incubation at room temperature, MTs in samples were spun through a 100 μl or 1 ml Cushion Buffer at 100,000 × *g* for 40 min in a TLA 120.1 or MLA-150 rotor (Beckman). Both supernatants and pellets were collected and analyzed.

**MT regrowth assay**. The MT regrowth assay was performed in RPE-1 cells. Briefly, RPE-1 cells were treated with nocodazole (2.5 μg/ml) for 2 h to depoly-merize MTs. Cells were then rinsed five times with ice-cold medium and moved to a dish with warm medium. MT regrowth was allowed in warm medium for indicated times. To detect only MT asters in Fig. 4d, cells were fixed with 4% paraformaldehyde at room temperature for 10 min. To detect centriolar protein localization during MT regrowth process, cells were fixed and permeabilized in −20 °C methanol for 2 min. Next, fixed cells were blocked with 3% NGS in 0.1% Triton X-100/PBS prior to incubation with primary antibodies.

**In vitro MT resolution assays**. GMPCPP "seed" MTs with incorporated Alexa-647 and biotin-labeled tubulin were attached to a neutravidin (Thermo Scientific, 31000)-coated coverslip. Alexa 561-labeled tubulin monomers were added and allowed to polymerize at room temperature for 5 min, then, 10 nM purified Flag-GFP-Vector or Flag-GFP-FSD1 proteins were added to the solution. The sample was imaged after reaction setup for 5 min on a total internal reflection fluorescence microscope (Olympus IX83-ZDC, UAPON OTIRF ×100 NA 1.49 oil objective) equipped with an Andor 897 Ultra EMCCD camera (Andor, UK). The raw images were analyzed with ImageJ (Fiji). For purified Flag-GFP-tagged proteins, Flag-GFP-tagged constructs were transfected into 293T cells, and then cells were harvested at 36 h after transfection and lysed with RIPA buffer containing 20 mM Tris-HCl, pH 7.5, 150 mM NaCl, 10 mM EDTA, 1% Triton X-100, 1% deoxycholate, 1 mM DTT, 0.5 mM PMSF, and phosSTOP. After sonication, the Flag M2 beads was used to immunoprecipitate the Flag-tagged proteins, then the immunoprecipitated Flag-GFP-tagged proteins were eluted with 3× Flag peptides (Sigma, F4799).

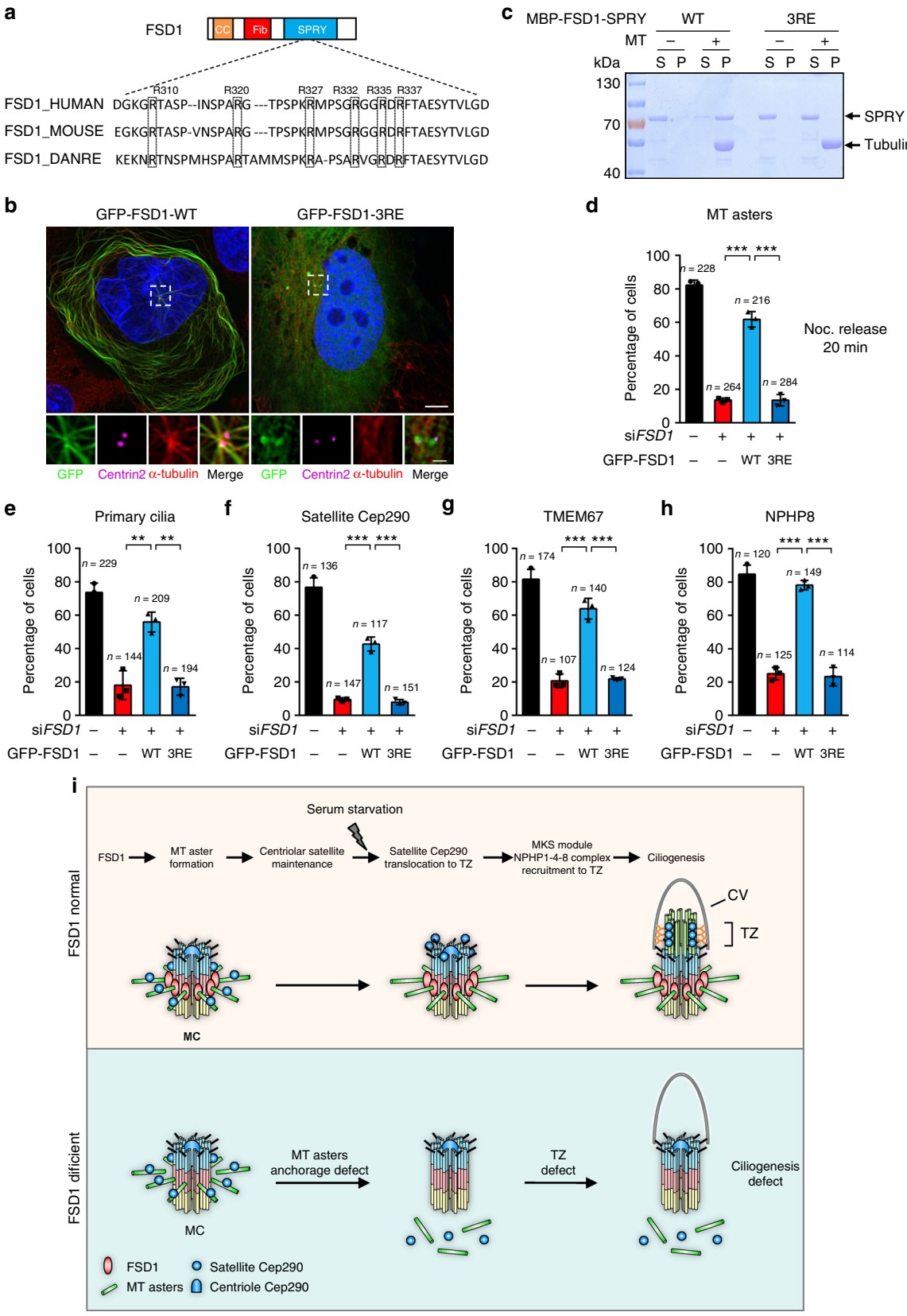

**Statistics**. Statistical calculations were performed with SPSS software. We tested data for normality and variance. All the data meet the normal distribution and statistical comparisons between two groups were carried out by two-tailed *t*-test. For all tests, differences were considered statistically significant if *P*-values were <0.05 (as indicated with *, *P* < 0.05; **P* < 0.01; ***P* < 0.001; NS, not significant). No statistical methods were used to predetermine sample size. The experiments were not randomized. No samples were excluded. The investigators were blinded to assess all the staining assays.

**Fig. 8** The microtubule-binding activity of FSD1 is essential for ciliogenesis. **a** A representative sequence of amino-acid alignment of the FSD1-SPRY domain was shown. The dotted box indicates six conserved arginine residues. **b** Expression of GFP-FSD1 WT, but not FSD1 3RE (R332E, R335E, and R337E) mutant, induced microtubule bundles. Scale bars, 5 μm (main image) and 1 μm (magnified region). **c** Mutation of three conserved arginine residues to glutamic acid (R332E, R335E, and R337E) abolishes the microtubule-binding activity of FSD1-SPRY domain. Purified FSD1-SPRY-WT, but not FSD1-SPRY-3RE, was co-pelleted with Taxol-stabilized microtubules. **d** Expression of GFP-FSD1 WT, but not 3RE mutant, rescued MT aster formation defects caused by FSD1 depletion. RPE-1 cells transfected with indicated siRNA and plasmids were subjected to a microtubule regrowth assay after nocodazole release 20 min. **e** Expression of GFP-FSD1 WT, but not 3RE mutant, rescued ciliogenesis defects caused by FSD1 depletion in quiescent cells. **f** Expression of GFP-FSD1 WT, but not 3RE mutant, rescued Cep290 localization at centriolar satellites in cycling cells. **g** Expression of GFP-FSD1 WT, but not 3RE mutant, rescued TMEM67 localization at centrioles in quiescent cells. **h** Expression of GFP-FSD1 WT, but not 3RE mutant, rescued NPHP8 localization at centrioles in quiescent cells. **i** Model of MT aster promoting ciliary transition zone assembly. FSD1 and other subdistal appendage proteins, such as Ninein, Kif3a, facilitate the formation of MT asters by anchoring MTs at the mother centriole. MT asters keep Cep290 at the centriolar satellites and promote transition zone assembly during ciliogenesis. MC mother centriole, MT microtubule, TZ transition zone, CV ciliary vesicle. Data are presented as mean ± s.d. of three independent experiments. n number of cells. In all panels, statistical comparisons between two groups were carried out by two-tailed t-test. **P < 0.01, ***P < 0.001

## Data availability
The data that support the findings of this study are available from the corresponding author upon reasonable request.

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

## Acknowledgements

We would like to thank Meng-Fu Bryan Tsou and Jianguo Chen for providing reagents. We thank Xin Liang for the assistance with the in vitro microtubule resolution assays. We thank Zhenggang Liu and Hongtao Yu for discussion and critical reading of the manuscript. We thank the Core Facilities of Life Sciences, Peking University and Technology Center for Protein Science, Tsinghua University for the assistance of super-resolution imaging and performing correlative light and electron microscopy strategy. This work was supported by Collaborative Innovation Center for Biotherapy. This work was also funded by the National Basic Research Program of China (2013CB910302 and 2014CB910603), National Natural Science Foundation of China (no. 81521064 and no. 81790252), and National key research and development program (2017YFC1601100; 2017YFC1601101; 2017YFC1601102; and 2017YFC1601104).

## Author contributions

H.-Y.L., X.-M.Z. and F.L. supervised the project; H.-Q.T. and H.-X.Q. designed and carried out most of the experiments; Z.-B.L. carried out most of the zebrafish experiment; Z.-Q.S., H.-B.H. and Y.-C.Z. contributed to the preparation of purified proteins; Y.H. and Y.-F.B. contributed to the preparation of complementary DNA vector constructs; Y.C., M.W. and G.W. carried out the statistics; Q.-Y.H., A.-L.L., T.Z., X.-M.Z. and H.-Y.L. analyzed the data; H.-Q.T., H.-X. Q., H.-Y.L. and X.-M.Z. wrote the paper. All authors discussed the results and commented on the manuscript.

## Additional information

**Competing interests:** The authors declare no competing interests.

