## [Peer Review File · Nature Communications]

Reviewers' Comments:

Reviewer #1:

Remarks to the Author:

In this study, Tu et al. have characterised FSD1, a centrosomal, microtubule-binding protein whose function was unknown. The authors have shown that FSD1 is localised to the subdistal region of both mother and daughter centrioles. Knockdown of FSD1 leads to ciliogenesis defects in both humans and zebrafish. They also have found that FSD1 depletion leads to compromised MT aster formation around the centrosome, which is attributed to failure of microtubule anchoring to the centrosome.

The data, in particular immunofluorescence images, are of high quality. However, having read this manuscript, I am wondering novelty of this work, which should justify publication in this journal. It is surely of interest that FSD1 is another centriolar protein that is required for ciliogenesis. The authors claim that in the abstract and the other places in the text "our study reveals an important function of MT asters in ciliogenesis, and mechanistically demonstrates the role of canonical cytoplasmic microtubules in specialized doublet microtubule assembly in cilia." However, the requirement of microtubule asters for ciliogenesis is already well established. Also, unfortunately, I cannot find any mechanistic understanding of the role of canonical cytoplasmic microtubules in specialized doublet microtubule assembly in cilia in this study at least in its present form.

Major points

- 1) FSD1 is localised to the subdistal region of the centriole, which is nicely shown. However, it is not clear as to how this specific localisation is determined and regulated. There are no biochemical data to show an interaction network between FSD1 and other centrosomal proteins, and detailed information on localisation dependency among these proteins is missing.
- 2) The authors have shown that FSD1 binds to microtubules in vitro and is required for microtubule anchoring to the centrosome and aster formation in vivo. There is a huge gap between these two results. Microtubule binding on its own does not explain the role of FSD1 in microtubule anchoring. To logically connect these two results, further data are necessary. I would expect that FSD1 specifically binds to the minus end of microtubules (the anchoring site) like γ -tubulin complex. As the authors have materials to address this critical point, TIRF assay should be performed, which would greatly increase the quality of this work.
- 3) The authors have shown that FSD1 knockdown leads to the disappearance of PCM1 from the centrosomal region. This is a very interesting observation if it is true. It is known that PCM1 does not disappear from the centrosomal region by the perturbation of microtubule structures; instead, PCM1 is dispersed away from the centrosome. I envisage that FSD1 regulates PCM1 localisation. It is even possible that the primary role for FSD1 is regulation of PCM1. In fact, all the data shown in this study could be explained by compromised PCM1 function. The authors should perform additional experiments to address this point.

Minor points

- 4) In several places, the authors insist that microtubule anchorage occurs at the subdistal appendage within the centriole. This description is not correct. It is known that other regions, such as the pericentriolar material (PCM), plays a critical role in microtubule anchoring.
- 5) In the introduction (no page numbers are shown in the text), it is described "suggesting a connection between MT aster anchoring and cilia assembly. Hence, there is growing interest in revealing whether and how MT asters are involved in ciliogenesis." These sentences do not represent the current knowledge in the field. In my view, the role of MT asters in ciliogenesis is firmly established, as centriolar satellites that are essential for ciliogenesis (analysed in this study) need MT asters for its proper localisation and function.

Errors

did not affected-did not affect

our result strongly suggest-our results strongly suggest

Reviewer #2:

Remarks to the Author:

In this manuscript, Tu et al reveals the role of FSD1 in anchoring microtubules (MTs) around the mother centriole and affecting centriolar satellite (CS) protein localization. This work identifies a novel player essential for CS positioning that is required for ciliogenesis. The authors used superresolution microscopy to show that FSD1 is a protein localized between subdistal appendage Ninein and proximal end Ninein, the region they termed subdistal region of the centriole. FSD1 knockdown (KD) reduces ciliation frequency at the stage of transition zone (TZ) assembly. They also found that FSD1 KD resulted in random orientations of MTs around the centrosome 20 min after nocodazole release, suggesting FSD1's role in MT anchoring. They showed FSD1 is essential for localization of PCM1 and Cep290 in CSs potentially through its function of MT anchoring. They found FSD1 directly binds to MT through its SPRY domain. Some TZ proteins are missing in FSD1 KD, potentially due to the recruiting failure of Cep290 from CS. Together, this study suggests that FSD1 is a centriolar subdistal region protein responsible for binding to MTs and tethering CSs, filling a knowledge gap in the centriole's role in CS interaction for proper ciliogenesis.

There are two concerns that require major efforts to revise the manuscript:

1. All fluorescent images seem to suggest that the mother centrioles the authors observed were long. For example, the epifluorescent images of first rows of Fig. 3b show that the distance between Cep162 and C-Nap1 is ~ 1 μm . Superresolution images from STED also show long centrioles. Fig. 3e shows the distance between Cep164 and FSD1 is ~ 500 nm, and Fig. 6b shows the distance between FSD1 and proximal Ninein is ~ 200 nm, making the total length of a mother centriole ~ 700 nm. In contrast, EM in Fig. 2f shows that the length of a mother centriole is ~ 500 nm, which is more typical for an RPE-1 centriole. The authors have to figure out how to address these discrepancies to convince the readers that all localization studies to define the position of FSD1 all reliable.

2. If indeed the localization is accurate, Fig. 6b shows that FSD1 is ~ 200 away from Ninein at the subdistal appendage level. This superresolution study actually illustrates that FSD1 does not belong to subdistal appendages, but instead localizes to the middle to proximal region of a centriole. Furthermore, FSD1 KD does not affect Ninein localization, and vice versa, showing that they are independent. Therefore, the authors should remove statement trying to categorize FSD1 as a subdistal region or a protein related to subdistal appendage functions. Subdistal region is somewhat misleading so the authors should avoid using it but instead stating it as correctly as their description of "narrow region" in the Discussion section (or use other terms such as "middle region"). Cartoons of Figs. 3f, 4g, 6b, and 8 should also be drawn in scale to the superresolution result of Fig. 6b to avoid confusion.

Minor comments:

1. One of the important elements to link the relationship between MT asters and ciliogenesis is that FSD1 plays a role in anchoring MT asters. The absence of MT asters after 20-min nocodazole release of FSD1 KD shown in Fig. 5d is the primary evidence for the FSD1's role in MT asters. Because of the importance of this image, the authors should provide multiple images like this in the Supplementary Materials to clearly illustrate the disappearance of MT asters after FSD1 KD.

2. The subtitle "MT Asters Anchored at Centrosomes Are Required for the Maintenance of Centriolar Satellites" does not accurately reflect the finding of that section, because that section

only suggests centriolar satellite proteins are transported on MT, not much about MT asters. The authors may consider changing it to something like "Localization of centriolar satellite proteins depends on dynamic MTs".

3. English throughout the manuscript should be edited where quite several mistakes can be found.

Reviewer #3:

Remarks to the Author:

In this well written manuscript the authors indicate a novel role for microtubule (MT) asters anchored by FSD1 for transition zone assembly during ciliogenesis. The authors showed that FSD1 knockdown inhibits primary cilia formation in vitro. FSD1 knockdown did not affect localization of proteins important for early steps of ciliogenesis. However, the group demonstrated that FSD1 was important for recruitment of several transition zone proteins. By using Stimulated Emission Depletion Microscopy (STED), the authors showed that FSD1 forms a ring structure at the subdistal region of the centrioles and does not localize to the transition zone. Furthermore, they showed that FSD1 is essential for localization of PCM1 and CEP290 at centriolar satellites (CS) and that CEP290 and PCM1 was transported by dynein along microtubules. The results indicate that FSD1 anchors MT asters to the centrosome and binds microtubules directly via its SPRY domain. They found that FSD1 inactivation disrupts MT aster anchoring and the dynamic Cep290 localization at the CS. This results in subsequent TZ assembly defects and, thus, indicates an important role of MT aster anchoring for ciliogenesis.

Overall, the in vitro studies are well done and conclusions are justified by the data. However, there are major and minor concerns that the group should address.

Major concerns

1. A morpholino (MO) zebrafish model has been used to assess *fsd1* function in zebrafish ciliogenesis. CRISPR/Cas9 tools have made genome editing widely possible and should thus be used to create mutants, which can be used to validate the MO by comparison. Furthermore, when creating MOs, distinct guidelines should be followed in order to distinguish specific phenotypes from potential off target effects of MOs (Stainier et al, PLoS Genet., 2017).
2. Authors did not provide information on the amount of MO injected. A dose response curve should be provided
3. The authors have used a multiple MO approach (ATG and splice blocking). *Fsd1* atg morpholino efficiency was validated by pEGFP- N1-*fsd1* plasmid (Supp. Fig. 2a). This control, however, is no longer recommended as suppression of GFP expression is generally observed and the suppression of the GFP expression does not test the effect of the MO on the endogenous RNA. The efficiency of the splice MO was shown by RT PCR and sequencing. The authors should add the sequencing data for the splice MO.
4. In Fig. 1 d, e the phenotype of a curved body and pericardial edema is shown. It should be mentioned, that these phenotypes are a rather unspecific MO effect and should not be assumed as ciliopathy associated symptoms. The Ctr-MO and misMO used in this experiment are only a control for developmental delay and cannot serve as a control for the specificity of the experimental MO (splice and ATG). The authors should state this in their article.
5. In Supp. Fig. 2c, d rescue experiments are shown. Zebrafish *fsd1* mRNA partially rescued the LR asymmetry defects induced by *fsd1* aMO. A control experiment, using mutant RNA should also be conducted. Furthermore, authors should state that the mRNA injected was lacking MO-binding sites.

Minor concerns

- The authors should add p values to figure 1 e, g and i.
- Please explain the abbreviation: KV (page 7 line 10).
- The authors should define the word "often" used on page 9 line 4. The previous sentence stated:

“In control cells more than 80% of mature centrioles docked onto vesicle membranes”. What is the percentage in FSD1 depleted cells?

- Supp. Fig 1 d, e, i. Figures should be shown at higher magnitude.
- Supp. Fig 1 j, p values missing.
- Figure legend Supp. Fig 6 a. The legend is hard to understand. Please correct the sentence.

Point to point responses to reviewer's comments:

Reviewer #1:

In this study, Tu et al. have characterised FSD1, a centrosomal, microtubule-binding protein whose function was unknown. The authors have shown that FSD1 is localised to the subdistal region of both mother and daughter centrioles. Knockdown of FSD1 leads to ciliogenesis defects in both humans and zebrafish. They also have found that FSD1 depletion leads to compromised MT aster formation around the centrosome, which is attributed to failure of microtubule anchoring to the centrosome.

The data, in particular immunofluorescence images, are of high quality. However, having read this manuscript, I am wondering novelty of this work, which should justify publication in this journal. It is surely of interest that FSD1 is another centriolar protein that is required for ciliogenesis. The authors claim that in the abstract and the other places in the text "our study reveals an important function of MT asters in ciliogenesis, and mechanistically demonstrates the role of canonical cytoplasmic microtubules in specialized doublet microtubule assembly in cilia." However, the requirement of microtubule asters for ciliogenesis is already well established. Also, unfortunately, I cannot find any mechanistic understanding of the role of canonical cytoplasmic microtubules in specialized doublet microtubule assembly in cilia in this study at least in its present form.

Response: We appreciate the reviewer's encouraging comments on our manuscript. The reviewer found our findings interesting and indicated that FSD1 as a centriolar protein plays an important role in ciliogenesis, and our data are of high quality. On the other hand, the reviewer suggested that our descriptions of the requirement of microtubule asters for ciliogenesis are inaccurate. We apologize for the confusion caused by our unclear descriptions. Following the reviewer's suggestion, we have re-written the manuscript to further clarify this issue.

According to previous studies, Ninein, Kif3a, EB1, and CEP19/CAP350/FOP proteins are required for MT aster anchoring (Bahareh A. Mojarad, et al. *Open Biology* 2017, 7 (6) :170114). All of these known MT aster anchoring proteins are also required for ciliogenesis (Ning Huang, et al. *Nature Communications*, 2017, 8 :15057). Although these previous reports are consistent with the notion that the status of MT aster anchoring is correlated with ciliogenesis, these studies did not address whether these MT anchoring proteins promotes ciliogenesis through their anchoring ability by creating mutants that specifically disrupt that activity. Several studies and reviews on MT anchoring proteins have raised this point (PAGE 8, line120, Bahareh A. Mojarad, et al. *Open Biology* 2017, 7 (6) :170114). Our data show that FSD1 is a new MT aster anchoring protein required for ciliogenesis. Importantly, an FSD1 mutant that fails to bind to microtubules and does not anchor MT aster also loses its activity to regulate ciliogenesis. Thus, these findings provide stronger evidence that MT aster is required for ciliogenesis. In addition, our subsequent data (*siFSD1*, *siNinein* and *siKif3a*, Fig. 5g, h) further suggest that MT asters specifically contribute to transition zone assembly in ciliogenesis. This finding is novel.

We have removed the speculation about the contribution of canonical cytoplasmic microtubules in cilia assembly in the revised manuscript.

This reviewer also provided some valuable suggestions for further improvement of our manuscript. We have performed additional experiments as suggested and have re-written the manuscript. As a result, the manuscript has been greatly improved. Below are responses to all the points raised by this reviewer.

Major points

1) FSD1 is localised to the subdistal region of the centriole, which is nicely shown. However, it is not clear as to how this specific localisation is determined and regulated. There are no biochemical data to show an interaction network between FSD1 and other centrosomal proteins, and detailed information on localisation dependency among these proteins is missing.

Response: Many thanks to the reviewer for this great suggestion. To investigate how the centriolar localization of FSD1 is determined and regulated, we first examined whether the localization of FSD1 could be affected by other MT anchorage proteins (a total of 12 proteins) (Fig. 7a). We had already shown that Ninein and Kif3a were not required for FSD1 localization at centrosomes in our original manuscript. We further tested other anchoring proteins, including p150, ODF2, PCM1, EB1, EB3, SXX2IP, TCHP and the CEP19/CAP350/FOP module. Interestingly, we found that both CAP350 and FOP are required for the localization of FSD1 on centrioles (Fig. 7a). FSD1 is also required for the localization of CAP350 on centrioles, but not FOP (Fig. 7b). CEP19 is dispensable for the localization of FSD1, and vice versa (Fig. 7a and 7b). We also observed the interaction between FSD1 and the CAP350-FOP complex (Fig. 7c and 7d). These new data indicate that the CAP350-FOP complex provides centrosomal docking sites for FSD1, which then anchors microtubules at centrosomes. FSD1 also contributes to the localization of CAP350 at centrosomes in a positive feedback. These results further indicate that FSD1 is the downstream target of several known anchoring proteins. We have added these new data as a new figure (Fig. 7) and rewritten the discussion in our revised manuscript.

2) The authors have shown that FSD1 binds to microtubules in vitro and is required for microtubule anchoring to the centrosome and aster formation in vivo. There is a huge gap between these two results. Microtubule binding on its own does not explain the role of FSD1 in microtubule anchoring. To logically connect these two results, further data are necessary. I would expect that FSD1 specifically binds to the minus end of microtubules (the anchoring site) like -tubulin complex. As the authors have materials to address this critical point, TIRF assay should be performed, which would greatly increase the quality of this work.

Response: This is a great point. We have now performed the TIRF assay, which indicates that FSD1 could bind to microtubule specifically in vitro (Fig. 6h). Interestingly, a small portion of GFP-FSD1 could directly bind to the microtubule minus ends (Fig. 6h). Thus, these results are more consistent with the fact that FSD1 binds to microtubules in vitro and is required for microtubule anchoring to the centrosome through its ability to bind microtubules (Fig. 6c-h and 8a-d). We have added this

critical data in our revised manuscript (Fig. 6h). We really appreciate the reviewer's constructive suggestion, which greatly increases the quality of this work.

3) The authors have shown that FSD1 knockdown leads to the disappearance of PCM1 from the centrosomal region. This is a very interesting observation if it is true. It is known that PCM1 does not disappear from the centrosomal region by the perturbation of microtubule structures; instead, PCM1 is dispersed away from the centrosome. I envisage that FSD1 regulates PCM1 localisation. It is even possible that the primary role for FSD1 is regulation of PCM1. In fact, all the data shown in this study could be explained by compromised PCM1 function. The authors should perform additional experiments to address this point.

Response: We appreciate the reviewer's suggestion. We performed additional experiments to test the possibility that "*the primary role for FSD1 is regulation of PCM1*". Our new data showed that unlike the knockdown of PCM1, the FSD1 knockdown did not affect the centriole localization of Ninein, Pericentrin and Centrin2 (Fig.6a and S6d), which are downstream proteins of PCM1 (Alexander Dammermann, et al. *Journal of Cell Biology*, 2002 , 159 (2) :255-266). These new results suggest that FSD1 might not directly regulate PCM1. Our original data have also shown that the FSD1-3RE mutant without MT-binding activity could not regulate MT aster anchorage and ciliogenesis. Our new data further show that the WT FSD1, but not 3RE mutant, could rescue the defect of PCM1 localization caused by FSD1 depletion (Fig. S9g). Thus, these data indicate that the critical role of FSD1 in ciliogenesis and in localization of CS protein (including PCM1) mainly depends on its MT anchoring ability.

We completely understand the concern of the referee about the PCM1 disappearance caused by FSD1 siRNA. We ourselves have also been confused by this phenomenon for a long time. We thus communicated with an expert, Dr. Simon Holst Bekker-Jensen (*University of Copenhagen, Copenhagen, Denmark*), who observed the disappearance of PCM1, but not the dispersion under UV light treatment. Dr. Bekker-Jensen told us "*centriolar satellite integrity is very sensitive to cell stress*" (Bine H Villumsen, et al. *The EMBO Journal* 2013, 32, 3029-3040). He also said that "*Besides classical stress treatments (UV light etc.), we often see that siRNA or plasmid transfection lead to depletion of satellites, as a pleiotropic effect of e.g. siRNA toxicity or transfection stress. It does not happen with every kind of siRNA transfection, but we see it regularly, especially with very toxic siRNAs*". He suspected that the PCM1 depletion may be an siRNA transfection stress response and suggested that "*You could try to treat your cells with an inhibitor against p38 or MK2 for a couple of hours before you fix them and see if the satellites come back.*"

According to his suggestions, we analyzed the possible contribution of siRNA toxicity or transfection stress on centriolar satellite with a p38 inhibitor. Our results show that the disappearance of PCM1 induced by FSD1 siRNA was indeed prevented by the p38 inhibitor-SB203580 (Fig. S6e). We now observe PCM1 dispersion under these conditions. Importantly, the loss of Cep290 induced by the knockdown of FSD1 was not affected by the p38 inhibitor (Fig. S6e). Thus, these results suggested that the disappearance of PCM1 induced by FSD1 siRNA might potentially be a p38-mediated transfection stress response. We have added this result and discussion in our revised manuscript.

Minor points

4) *In several places, the authors insist that microtubule anchorage occurs at the subdistal appendage within the centriole. This description is not correct. It is known that other regions, such as the pericentriolar material (PCM), plays a critical role in microtubule anchoring.*

Response: We agree with the reviewer and apologize for the inaccurate description. It is known that the subdistal region within the centriole acts as a major microtubule anchorage site, and other regions, such as the pericentriolar material (PCM), also play critical roles in microtubule anchoring. We have clarified the contribution of the pericentriolar material (PCM) in MT anchoring in the Introduction section of our revised manuscript (Page 3, Line 18).

5) *In the introduction (no page numbers are shown in the text), it is described “suggesting a connection between MT aster anchoring and cilia assembly. Hence, there is growing interest in revealing whether and how MT asters are involved in ciliogenesis.” These sentences do not represent the current knowledge in the field. In my view, the role of MT asters in ciliogenesis is firmly established, as centriolar satellites that are essential for ciliogenesis (analysed in this study) need MT asters for its proper localisation and function.*

Response: We apologize for the confusion caused by our unclear descriptions. Following the reviewer’s suggestion, we have re-written the manuscript to further clarify this issue (Page 3, Line 24). We hope to explain our points of view in our original manuscript. According to the previous studies, centriolar satellite proteins such as Cep290 and PCM1 are required for ciliogenesis (Joon Kim, et al, *Human Molecular Genetics*, 2008, 17 (23) :3796-3805). Meanwhile, the localization of centriolar satellite (CS) proteins required polymerized MTs (Alexander Dammermann, et al. *Journal of Cell Biology*, 2002, 159 (2) :255-266). Thus, these previous reports are indeed consistent with the notion that MT aster might be also required for ciliogenesis. These studies did not, however, show that polymerized MTs promotes ciliogenesis through centriolar satellite proteins. Moreover, these studies did not test whether MT aster anchoring around centrosomes specifically mediates this process. In our study, knockdowns of two key MT aster anchoring proteins, including Ninein and Kif3a, disrupted Cep290 location at CS. Our data also show that FSD1 is a new MT aster anchoring protein that is required for CS protein proper localization and function. Importantly, FSD1 mutant that fails to bind to microtubules and cannot anchor MT asters also loses its ability to regulate the localization of CS proteins and ciliogenesis. Thus, these data provide stronger evidence that MT aster is required for CS location and ciliogenesis. Our subsequent data (*siFSD1*, *siNinein* and *siKif3a*, Fig. 5g, h) further suggest that MT asters might specifically contribute to transition zone assembly in ciliogenesis. This finding is novel.

Errors:

*did not affected-did not affect;
our result strongly suggest-our results strongly suggest*

Response: We have corrected these errors in our revised manuscript.

Reviewer #2 (Remarks to the Author):

In this manuscript, Tu et al reveals the role of FSD1 in anchoring microtubules (MTs) around the mother centriole and affecting centriolar satellite (CS) protein localization. This work identifies a novel player essential for CS positioning that is required for ciliogenesis. The authors used superresolution microscopy to show that FSD1 is a protein localized between subdistal appendage Ninein and proximal end Ninein, the region they termed subdistal region of the centriole. FSD1 knockdown (KD) reduces ciliation frequency at the stage of transition zone (TZ) assembly. They also found that FSD1 KD resulted in random orientations of MTs around the centrosome 20 min after nocodazole release, suggesting FSD1's role in MT anchoring. They showed FSD1 is essential for localization of PCMI and Cep290 in CSs potentially through its function of MT anchoring. They found FSD1 directly binds to MT through its SPRY domain. Some TZ proteins are missing in FSD1 KD, potentially due to the recruiting failure of Cep290 from CS. Together, this study suggests that FSD1 is a centriolar subdistal region protein responsible for binding to MTs and tethering CSs, filling a knowledge gap in the centriole's role in CS interaction for proper ciliogenesis.

Response: We thank the reviewer for the cogent summary of the major contributions of our study. The reviewer pointed out that “*FSD1 is a centriolar subdistal region protein responsible for binding to MTs and tethering CSs, filling a knowledge gap in the centriole's role in CS interaction for proper ciliogenesis.*” Furthermore, this reviewer also provided some constructive suggestions for further improvement of our manuscript. We have performed additional experiments as suggested. As a result, the manuscript has been greatly improved. Below are our responses to the points raised by this reviewer.

There are two concerns that require major efforts to revise the manuscript:

1. All fluorescent images seem to suggest that the mother centrioles the authors observed were long. For example, the epifluorescent images of first rows of Fig. 3b show that the distance between Cep162 and C-Nap1 is ~1 μ m. Superresolution images from STED also show long centrioles. Fig. 3e shows the distance between Cep164 and FSD1 is ~500 nm, and Fig. 6b shows the distance between FSD1 and proximal Ninein is ~200 nm, making the total length of a mother centriole ~700 nm. In contrast, EM in Fig. 2f shows that the length of a mother centriole is ~500 nm, which is more typical for an RPE-1 centriole. The authors have to figure out how to address these discrepancies to convince the readers that all localization studies to define the position of FSD1 all reliable.

Response: Many thanks to the reviewer for his/her keen observations. To address this concern, we calculated the distance between Cep162 and C-Nap1 in Fig. 3b more accurately. Briefly, we drew a line along axial axis between Cep162 and C-Nap1, and measured their fluorescence profile plots (see below). The distance between Cep162 and C-Nap1 was calculated as the distance between their corresponding Gaussian peaks. As shown in Fig. 3b, the distance between Cep162 and C-Nap1 is ~600 nm (line1 and line 2). Compared to EM, fluorescence imaging has lower resolution. The epifluorescent images of Fig. 3b in our manuscript are taken by DeltaVision at standard resolution of light microscopy, and the pixel size of this image is 104 nm, which leads to the length of centrioles to fluctuate in the range of 600 \pm 104 nm. According to the above method, we further calculated the

distance in the super-resolution STED images in Fig. 3e and 6b. Fig. 3e showed the distance between FSD1 and Cep164 is ~ 400 nm (line 3), Fig. 6b showed the distance between FSD1 and Ninein is ~ 200 nm (line 4), making the total length of a mother centriole to be ~ 600 nm. The images of Fig. 3e and 6b in our manuscript are taken by STED, and the resolution of STED is about 50 nm, which leads to the length of centrioles to fluctuate in the range of 600 ± 50 nm. In addition, during STED experiments, the light often scans from different angles of two centrioles to offer a composite view, which could also lead to the length of centrioles to fluctuate. Taken together, the factor mentioned above might explain why the STED results are not as accurate as the EM results.

Fig. 3b

Fig. 3e

Fig. 6b

2. If indeed the localization is accurate, Fig. 6b shows that FSD1 is ~ 200 away from Ninein at the subdistal appendage level. This superresolution study actually illustrates that FSD1 does not belong to subdistal appendages, but instead localizes to the middle to proximal region of a centriole. Furthermore, FSD1 KD does not affect Ninein localization, and vice versa, showing that they are independent. Therefore, the authors should remove statement trying to categorize FSD1 as a

subdistal region or a protein related to subdistal appendage functions. Subdistal region is somewhat misleading so the authors should avoid using it but instead stating it as correctly as their description of “narrow region” in the Discussion section (or use other terms such as “middle region”). Cartoons of Figs. 3f, 4g, 6b, and 8 should also be drawn in scale to the superresolution result of Fig. 6b to avoid confusion.

Response: Many thanks to the reviewer for his/her keen observations and suggestions. We agree entirely with the reviewer on these points and have revised the description of FSD1 location as the “middle region” beneath subdistal appendages of the centriole. According to the reviewer’s suggestion, we have drawn the corresponding cartoons (Figs. 3f, 4g, 6b, 8i) in scale to the result of Fig. 6b in our revised manuscript.

Minor comments:

1. One of the important elements to link the relationship between MT asters and ciliogenesis is that FSD1 plays a role in anchoring MT asters. The absence of MT asters after 20-min nocodazole release of FSD1 KD shown in Fig. 5d is the primary evidence for the FSD1’s role in MT asters. Because of the importance of this image, the authors should provide multiple images like this in the Supplementary Materials to clearly illustrate the disappearance of MT asters after FSD1 KD.

Response: We appreciate the reviewer’s suggestion. We have provided multiple images of MT asters in FSD1 KD cells in Fig. S8a to show that FSD1 plays a role in anchoring MT asters in our revised manuscript.

2. The subtitle “MT Asters Anchored at Centrosomes Are Required for the Maintenance of Centriolar Satellites” does not accurately reflect the finding of that section, because that section only suggests centriolar satellite proteins are transported on MT, not much about MT asters. The authors may consider changing it to something like “Localization of centriolar satellite proteins depends on dynamic MTs”.

Response: Many thanks to the reviewer’s great suggestion. We have changed the subtitle according to his/her advice in our revised manuscript.

3. English throughout the manuscript should be edited where quite several mistakes can be found.

Response: We have tried our best to identify and correct errors in the text.

Reviewer #3 (Remarks to the Author):

In this well written manuscript the authors indicate a novel role for microtubule (MT) asters anchored by FSD1 for transition zone assembly during ciliogenesis. The authors showed that FSD1 knockdown inhibits primary cilia formation in vitro. FSD1 knockdown did not affect localization of proteins important for early steps of ciliogenesis. However, the group demonstrated that FSD1 was important for recruitment of several transition zone proteins. By using Stimulated Emission

Depletion Microscopy (STED), the authors showed that FSD1 forms a ring structure at the subdistal region of the centrioles and does not localize to the transition zone. Furthermore, they showed that FSD1 is essential for localization of PCMI and CEP290 at centriolar satellites (CS) and that CEP290 and PCMI was transported by dynein along microtubules. The results indicate that FSD1 anchors MT asters to the centrosome and binds microtubules directly via its SPRY domain. They found that FSD1 inactivation disrupts MT aster anchoring and the dynamic Cep290 localization at the CS. This results in subsequent TZ assembly defects and, thus, indicates an important role of MT aster anchoring for ciliogenesis.

Overall, the in vitro studies are well done and conclusions are justified by the data. However, there are major and minor concerns that the group should address.

Response: We thank the reviewer for the cogent summary of the major contributions of our study and for the overall positive assessment. According to reviewer's suggestions, we have performed a number of in vivo experiments to improve the quality of our manuscript, including rescue experiments, a dose response curve assay, as well as generating a new *fsd1* maternal-zygotic mutant to confirm the phenotypes. The detailed point-by-point responses to the reviewer's comments are provided below.

Major concerns

1. A morpholino (MO) zebrafish model has been used to assess fsd1 function in zebrafish ciliogenesis. CRISPR/Cas9 tools have made genome editing widely possible and should thus be used to create mutants, which can be used to validate the MO by comparison. Furthermore, when creating MOs, distinct guidelines should be followed in order to distinguish specific phenotypes from potential off target effects of MOs (Stainier et al, PLoS Genet., 2017).

Response: We thank this reviewer for these suggestions. Actually, after submitting our manuscript, we started to create *fsd1* mutant zebrafish by CRISPR/Cas9. We have now finished the experiments and obtained results similar to those of the MO zebrafish experiments. As shown in Fig. 1k and S3, these results further strengthen our conclusion that *fsd1* plays an important role in proper ciliogenesis and ciliary function during embryonic development, including ciliogenesis in KV, curved body and LR asymmetry. Following to the reference cited by the reviewer, we next clarified the specific phenotypes of FSD1 in subsequent experiments (see below).

2. Authors did not provide information on the amount of MO injected. A dose response curve should be provided.

Response: We thank the reviewer for pointing out this important issue. The amount of *fsd1* aMO, sMO and misMO injected in each embryo was 1.5, 3.5 and 1.5 ng/embryo, respectively. A dose-dependent response assay was performed, in which the phenotypes including ciliogenesis defects in KV, LR asymmetry defects and curved body were characterized. We have added these data in Fig 1 d-j and S2c in our revised manuscript.

3. The authors have used a multiple MO approach (ATG and splice blocking). *Fsd1 atg* morpholino efficiency was validated by pEGFP- N1-*fsd1* plasmid (Supp. Fig. 2a). This control, however, is no longer recommended as suppression of GFP expression is generally observed and the suppression of the GFP expression does not test the effect of the MO on the endogenous RNA. The efficiency of the splice MO was shown by RT PCR and sequencing. The authors should add the sequencing data for the splice MO.

Response: We appreciate the reviewer's suggestion. To further confirm the efficiency of *fsd1 atg* morpholino, we extracted proteins from control and *fsd1* aMO-injected embryos and then detected the endogenous *fsd1* expression at 26 hpf by western blotting. As shown in Fig. S2a, endogenous Fsd1 was obviously decreased in *fsd1* aMO- injected embryos. The cDNA sequence of *fsd1* sMO-injected embryos by sequencing showed that the exon2 skipping occurred in *fsd1* sMO-injected embryos, but not in WT embryos (see Fig. S2b).|

4. In Fig. 1 d, e the phenotype of a curved body and pericardial edema is shown. It should be mentioned, that these phenotypes are a rather unspecific MO effect and should not be assumed as ciliopathy associated symptoms. The Ctr-MO and misMO used in this experiment are only a control for developmental delay and cannot serve as a control for the specificity of the experimental MO (splice and ATG). The authors should state this in their article.

Response: We appreciate the reviewer's suggestion. In our original Fig. 1d, e, *fsd1* misMO and aMO with 5-nucleotide difference were used at the same dose of 1.5 ng per embryos. The results showed that curved body was observed in *fsd1* aMO-injected embryos, but not in *fsd1* misMO-injected embryos, indicating that the *fsd1* misMO does not induce toxic effects during zebrafish development. Furthermore, in our revised manuscript, we injected the *fsd1* misMO at a higher dose (7 ng per embryo in Fig. 1d, e), but the curved body phenotype still did not occur. The curved body phenotype showed a dose response curve for the *fsd1* MOs. In addition, the specificities of *fsd1* aMO and sMO were validated by western blotting and RT-PCR/sequencing, respectively. Thus, these data further confirmed that the curved body was a result from *fsd1* gene depletion. Furthermore, the curved body phenotype is often observed in other morphants or mutants of cilia-related genes, such as *Cep162*, *miR129-3p* (Won-Jing Wang, et al. *Nature Cell Biology*, 2013, 15 (6):591-601; Jingli Cao, et al. *Nature Cell Biology*, 2012, 14 (7) :697-706). Taken together, our results indicate that the curved body phenotype in *fsd1* morphants might be specific to the *fsd1* gene.

5. In Supp. Fig. 2c, d rescue experiments are shown. Zebrafish *fsd1* mRNA partially rescued the LR asymmetry defects induced by *fsd1* aMO. A control experiment, using mutant RNA should also be conducted. Furthermore, authors should state that the mRNA injected was lacking MO-binding sites.

Response: We thank this reviewer for these thoughtful suggestions and apologize for our unclear descriptions. In our original rescue experiment, zebrafish *fsd1* mRNA (*fsd1* re-zmRNA) was resistant to *fsd1* aMO. We have added this detailed information in Fig. S2d. According to reviewer's suggestion, we synthesized a zebrafish *fsd1* mutant mRNA (*fsd1*-CC re-zmRNA), which lacks the *fsd1* aMO target sequences and encodes a truncated *fsd1* protein without the conserved SPRY and FN3 domains (see Fig. S2g). The result showed that *fsd1* mutant mRNA (*fsd1*-CC re-zmRNA) could

not rescue the LR asymmetry defects in *fsd1* morphants (see Fig. S2h, i). Collectively, the LR asymmetry defects occurred in *fsd1* morphants could be rescued by *fsd1* re-zmRNA (wild type, see Fig. S2e, f), but not *fsd1*-CC re-zmRNA, suggesting that *fsd1* is required for the LR asymmetry in zebrafish development.

Minor concerns

- The authors should add *p* values to figure 1 e, g and i.
- Please explain the abbreviation: KV (page 7 line 10).

Response: According to the reviewer's suggestion, we performed statistical analysis on the data of the original Fig. 1e, g and i and added the *p* values to these data. We also explained the meaning of KV (Kuffer's Vesicle) in our revised manuscript.

The authors should define the word "often" used on page 9 line 4. The previous sentence stated: "In control cells more than 80% of mature centrioles docked onto vesicle membranes". What is the percentage in FSD1 depleted cells?

Response: Many thanks to the reviewer's suggestion. In FSD1 depleted cells, about 90% (28/30) of mature centrioles docked onto vesicle membranes. We have added this percentage in our revised manuscript.

- *Supp. Fig 1 d, e, i. Figures should be shown at higher magnitude.*
- *Supp. Fig 1 j, p values missing.*
- *Figure legend Supp. Fig 6 a. The legend is hard to understand. Please correct the sentence.*

Response: Many thanks to the reviewer's suggestions.

1. We have replaced the original Fig. S1d, e, i. with higher magnitude ones in our revised manuscript.
2. We have added *p* values to Fig. S1j in our revised manuscript.
3. We have revised the legend of Fig. S7a in our revised manuscript.

Reviewers' Comments:

Reviewer #1:

Remarks to the Author:

In this revised manuscript, the authors addressed all my points raised in my previous evaluation. Therefore, as far as I am concerned, I am happy with this work.

One outstanding point:

I am slightly confused of the data shown in Supplementary Fig. 6d. In this figure, the authors show that signal intensities of two centrosomal/centriolar proteins (Centrin2 and Pericentrin), which were previously claimed to be cargo proteins transported to the centrosome/centriole via PCM1-dependent centriolar satellites (CS) (Dammermann et al., 2002), have not been decreased upon knockdown of FSD1, unlike that of PCM1. As FSD1 knockdown leads to the disappearance of PCM1 from CS (Fig. 4c), the impact on localisation of Centrin2 and Pericentrin derived from FSD1 or PCM1 knockdown should be similar. I will not request further experiments with regards to this issue, but this discrepancy should be acknowledged and I recommend the authors adding some explanations/comments on this result in the text.

Minor points to be corrected:

1) page 3, ninein

Ninein

2) page 4, independent of other anchoring proteins

This notion is misleading or incorrect, as FSD1 localisation to the centrosome/centriole is dependent upon the CAP350-FOP complex (Fig. 7a).

3) page 18, RPE

RPE-1

4) page 21, maintain

maintains

5) Zebrafish FSD1/fsd1

In many places that describe zebrafish *fsd1*, *fsd1* (FSD1) should be shown in the Italic style, not Roman, as it represents gene, rather than protein.

6) protein size markers

It is now normal practice to show the positions of size markers in immunoblotting images or SDS-PAGE gels.

Reviewer #2:

Remarks to the Author:

The response was satisfactory and the contribution on the role of FSD1 is novel. I have no further question about the manuscript.

Reviewer #3:

Remarks to the Author:

The authors have addressed all issues and substantially improved the manuscript.

Responses to Reviewers:

Reviewer #1:

In this revised manuscript, the authors addressed all my points raised in my previous evaluation. Therefore, as far as I am concerned, I am happy with this work.

One outstanding point:

I am slightly confused of the data shown in Supplementary Fig. 6d. In this figure, the authors show that signal intensities of two centrosomal/centriolar proteins (Centrin2 and Pericentrin), which were previously claimed to be cargo proteins transported to the centrosome/centriole via PCM1-dependent centriolar satellites (CS) (Dammermann et al., 2002), have not been decreased upon knockdown of FSD1, unlike that of PCM1. As FSD1 knockdown leads to the disappearance of PCM1 from CS (Fig. 4c), the impact on localisation of Centrin2 and Pericentrin derived from FSD1 or PCM1 knockdown should be similar. I will not request further experiments with regards to this issue, but this discrepancy should be acknowledged and I recommend the authors adding some explanations/comments on this result in the text.

Response: We thank the reviewer for carefully reading our revised manuscript and for these positive comments. As suggested by the reviewer, we also wondered why the effect of FSD1 knockdown on centrin2/pericentrin recruitment to centrosomes was not similar to PCM1 knockdown. We hypothesize that PCM1 might exist in another form, but they could not be easily detected after FSD1 depletion by our immunofluorescence assay, as the protein level of PCM1 examined here was not affected by FSD1 depletion. We consider that FSD1 depletion does not completely inactivate PCM1. We have now included these discussions in our revised manuscript (page 13).

Minor points to be corrected:

1) page 3, ninein

Ninein

2) page 4, independent of other anchoring proteins

This notion is misleading or incorrect, as FSD1 localisation to the centrosome/centriole is dependent upon the CAP350-FOP complex (Fig. 7a).

3) page 18, RPE

RPE-1

4) page 21, maintain

maintains

5) Zebrafish *FSD1/fsd1*

In many places that describe zebrafish fsd1, fsd1 (FSD1) should be shown in the Italic style, not Roman, as it represents gene, rather than protein.

6) protein size markers

It is now normal practice to show the positions of size markers in immunoblotting images or SDS-PAGE gels.

Response: We apologize for these mistakes in our text. Following the reviewer's suggestions, we have corrected these errors in our revised manuscript and labeled the protein size markers in our revised Figures.

Reviewer #2:

The response was satisfactory and the contribution on the role of FSD1 is novel. I have no further question about the manuscript.

Response: Many thanks to the reviewer's kind consideration and remarks on our paper that helped a lot to improve our manuscript.

Reviewer #3:

The authors have addressed all issues and substantially improved the manuscript.

Response: We appreciate the reviewer's sound view on our work and thank her/him again for the valuable suggestions improving our manuscript.